# Development of a large-eddy simulation subgrid model based on artificial neural networks: a case study of turbulent channel flow

Robin Stoffer[1], Caspar M. van Leeuwen[2], Damian Podareanu[2], Valeriu Codreanu[2], Menno A. Veerman[1], Martin Janssens[1], Oscar K. Hartogensis[1], and Chiel C. van Heerwaarden[1]

[1]Meteorology and Air Quality Group, Wageningen University, Wageningen, The Netherlands
[2]SURFsara, Amsterdam, The Netherlands

**Correspondence:** Robin Stoffer (robin.stoffer@wur.nl)

**Abstract.** Atmospheric boundary layers and other wall-bounded flows are often simulated with the large-eddy simulation (LES) technique, which relies on subgrid-scale (SGS) models to parameterize the smallest scales. These SGS models often make strong simplifying assumptions. Also, they tend to interact with the discretization errors introduced by the popular LES approach where a staggered finite-volume grid acts as an implicit filter. We therefore developed an alternative LES SGS model based on artificial neural networks (ANNs) for the computational fluid dynamics code MicroHH (v2.0). We used a turbulent channel flow (with friction Reynolds number $Re_\tau = 590$) as a test case. The developed SGS model has been designed to compensate for both the unresolved physics and instantaneous spatial discretization errors introduced by the staggered finite-volume grid. We trained the ANNs based on instantaneous flow fields from a direct numerical simulation (DNS) of the selected channel flow. In general, we found excellent agreement between the ANN predicted SGS fluxes and the SGS fluxes derived from DNS for flow fields not used during training. In addition, we demonstrate that our ANN SGS model generalizes well towards other coarse horizontal resolutions, especially when these resolutions are located within the range of the training data. This shows that ANNs have potential to construct highly accurate SGS models that compensate for spatial discretization errors. We do highlight and discuss one important challenge still remaining before this potential can be successfully leveraged in actual LES simulations: we observed an artificial build-up of turbulence kinetic energy when we directly incorporated our ANN SGS model into a LES simulation of the selected channel flow, eventually resulting in numeric instability. We hypothesize that error accumulation and aliasing errors are both important contributors to the observed instability. We finally make several suggestions for future research that may alleviate the observed instability.

## 1 Introduction

Large-eddy simulation (LES) is an often used technique to simulate turbulent atmospheric boundary layers (ABLs) and other wall-bounded geophysical flows with high Reynolds numbers (e.g. rivers). These turbulent flows are challenging to simulate because of their strong non-linear dynamics and large ranges of involved spatial and temporal scales. LES explicitly resolves only

the largest, most energetic turbulent structures in these flows, while parameterizing the smaller ones with so-called subgrid-scale (SGS) models. This allows LES to keep the total computational effort feasible for today's high-performance computing systems, but makes the quality of the results strongly dependent on the chosen SGS model. As an SGS model based on physical principles alone does not exist, the SGS models used today typically rely on simplifying assumptions in combination with ad-hoc empirical corrections (e.g Pope, 2001; Sagaut, 2006).

To briefly illustrate the effects simplifying assumptions can have, we take as an example the eddy-viscosity assumption used in the popular Smagorinsky model (Smagorinsky, 1963; Lilly, 1967) and several other SGS models. Crucially, the eddy-viscosity assumption introduces an alignment between the Reynolds stress and strain rate tensor that has not been verified in experimental data (Schmitt, 2007). This makes it impossible to produce both the correct Reynolds stresses and dissipation rates (Jimenez and Moser, 2000). As a consequence, eddy-viscosity SGS models often require ad-hoc manual corrections (e.g. tuning the Smagorinsky coefficient and/or implementing a wall-damping function) or multiple computationally expensive spatial filtering operations (e.g. scale-dependent dynamical Smagorinsky models (Bou-Zeid et al., 2005)) to achieve satisfactory results.

Data-driven machine learning techniques are, in contrast, much more flexible regarding their functional form, and thus may potentially help to circumvent the need for many of these simplifying assumptions. This is especially valid for artificial neural networks (ANNs): simple feed-forward ANNs with just one hidden layer are theoretically able to represent any continuous function on finite domains (i.e they are universal approximators; Hornik et al. (1989)).

A wide effort is therefore currently underway to explore the potential for ANNs and other machine learning techniques in flow and turbulence modelling (Brunton et al., 2020; Kutz, 2017; Duraisamy et al., 2019). In particular, multiple studies successfully modelled turbulence in Reynolds-Averaged Navier-Stokes (RANS) codes with machine learning techniques trained on high-fidelity direct numerical simulations (DNS) that resolve all relevant turbulence scales (e.g. Kaandorp and Dwight, 2020; Ling et al., 2016a, b; Wang et al., 2017; Wu et al., 2018; Singh et al., 2017).

Several other efforts in literature experimented with comparable approaches in both LES SGS modelling (Beck et al., 2019; Cheng et al., 2019; Gamahara and Hattori, 2017; Maulik et al., 2019; Milano and Koumoutsakos, 2002; Sarghini et al., 2003; Vollant et al., 2017; Wang et al., 2018; Xie et al., 2019; Yang et al., 2019; Zhou et al., 2019) and, interestingly, parameterizations in climate/ocean modelling (e.g. Bolton and Zanna, 2019; Brenowitz and Bretherton, 2019; Rasp, 2020; Yuval and O'Gorman, 2020). The studies focusing on LES SGS modelling, similarly used DNS fields as a basis, and subsequently applied a downscaling procedure to generate consistent pairs of coarse-grained fields (that are assumed to represent the fields that a LES code would generate) and the quantity of interest (e.g. the 'true' subgrid transport or the closure term itself). These pairs were then typically used to train ANNs in a supervised way. Some studies showed very promising results with this method, both in so-called *a priori, offline* tests (where the predicted quantity is directly compared to the ones derived from DNS) and so-called *a posteriori, online* tests (where the trained ANN is directly incorporated as a SGS model into a LES simulation). However, these studies mostly focused on 2d/3d (in)compressible isotropic turbulence (Beck et al., 2019; Guan et al., 2021; Maulik et al., 2019; Vollant et al., 2017; Wang et al., 2018; Xie et al., 2019; Zhou et al., 2019), and thus do not represent wall-bounded geoscientific flows. Furthermore, some of these studies (Beck et al., 2019; Maulik et al., 2019; Zhou

et al., 2019) resorted to ad-hoc adjustments (e.g. artificially introducing dissipation by combining with the Smagorinsky SGS model, neglecting all backscatter) to achieve stable a posteriori results. Such ad-hoc adjusments are not ideally preferred: they obscure the link between the a priori and a posteriori implementation, and re-introduce part of the assumptions that are ideally circumvented by using ANN SGS models.

There are also studies that attempted similar methods in cases that better represent ABLs. Some of them focused on LES wall modelling specifically (Milano and Koumoutsakos, 2002; Yang et al., 2019), which is challenging on its own because of the many unresolved near-wall motions that the wall model has to take into account. Sarghini et al. (2003) and Gamahara and Hattori (2017), in turn, focused on SGS modelling in the whole turbulent channel flow. Sarghini et al. (2003) used neural networks to predict the Smagorinsky coefficient in the Smagorinsky-Bardina SGS model (Bardina et al., 1980) reaching a computational time saving of about 20%. Gamahara and Hattori (2017) directly predicted the SGS turbulent transport with a neural network, using DNS during training. They got reasonable a priori results, but did not perform an a posteriori test. Another important step towards application of these methods in realistic atmospheric boundary layers was taken by Cheng et al. (2019). They performed an extensive a priori test for an ANN-based LES SGS model covering a wide range of grid resolutions and flow stabilities (from neutral channel flow to very unstable convective boundary layers). We emphasize though that for successful integration of ANN-based SGS models in practical applications, accurate and numerically stable a posteriori results are an important requirement. Recently, Park and Choi (2021) took a step in this direction by testing an ANN-based SGS model in a neutral channel flow both a priori and a posteriori. They found that their SGS model introduced numeric instability a posteriori, except when they neglected all back-scatter or only used single-point, rather than multi-point, inputs. However, selecting only single-point inputs, in turn clearly reduced the a priori performance. Hence, it remains an open issue whether and how the often observed high a priori potential of ANN SGS models, can be successfully leveraged in an a posteriori test, in particular for wall-bounded flows like ABLs.

In addition, all the previously mentioned ANN LES SGS models, together with traditional eddy-viscosity models, do not directly reflect the LES approach where a staggered finite-volume numerical scheme acts as an implicit filter, despite being a common practice when simulating ABLs. Traditional eddy-viscosity models are typically derived based on a generic filtering operation that does not consider the finite discrete nature of the used numerical grid (i.e. it is usually thought of as an analytical filter like a continuous top-hat filter), while the ANN SGS models so far did not attempt to compensate for all the discretization errors arising in simulations with staggered finite-volumes. These discretization errors, however, can strongly influence the resolved dynamics (e.g. Ghosal, 1996; Chow and Moin, 2003; Giacomini and Giometto, 2020), especially at the smallest resolved scales. Since the ANNs have access to both the instantaneous DNS flow fields and corresponding coarse-grained field during training, an unique opportunity arises to compensate also for instantaneous discretization errors in ANN SGS models.

Within this context, there have been a couple of noteworthy studies (Langford and Moser, 1999; Völker et al., 2002; Zandonade et al., 2004) that introduced the framework of *perfect* and *optimal* LES. Based on this framework, these studies approximated the full LES closure terms (that account for both the unresolved physics and all instantaneous discretization errors) with a data-driven approach based on DNS. The statistical method they used for this purpose though (i.e. stochastic estimation), still made additional assumptions about the functional form of the LES SGS model (e.g. linearity). A recent study by Beck

et al. (2019) therefore used ANNs to approximate, in a similar way as the aforementioned studies, the full LES closure terms. To construct based on these ANNs an LES SGS model that is numerically stable a posteriori, they combined the ANNs with eddy-viscosity models. They did not specifically focus on the discretization errrors associated with staggered finite-volume grids, and did not consider wall-bounded turbulent flows like ABLs.

In this study, we therefore made a first attempt to construct, based on DNS fields, an ANN SGS model that compensates for both the unresolved physics and the instantaneous discretization errors introduced by staggered finite-volume grids. Our ambition in doing so, is to eventually improve the a posteriori accuracy compared to LES with traditional SGS models like Smagorinsky. This may potentially reduce the computational cost involved in LES as well, as accurate results may still be achieved with much coarser, computationally cheaper resolutions than currently used.

To make a step towards this ambition, our aim with the current manuscript is two-fold:

1. Describe the framework of our ANN SGS model, which takes both the unresolved physics and instantaneous spatial discretization errors in finite-volume LES into account. This includes its theoretical foundations (Sect 2) and its implementation (Sect 3).

2. Characterize both the a priori and a posteriori performance of our ANN SGS model for a wall-bounded turbulent neutral channel flow, without resorting to previously used ad-hoc adjustments (Sect. 4). This includes a discussion about the numeric instability we observed a posteriori (Sect. 4.2), together with suggestions for future studies that may help to overcome the observed instability without needing the previously used ad-hoc adjustments (Sect. 5).

## 2 Theoretical framework ANN SGS model finite-volume LES

As mentioned in Sect. 1, one of our key objectives is to construct an ANN LES SGS model that compensates for the instantaneous discretization errors introduced by implicit filtering with staggered finite-volume numerical schemes. To derive such an SGS model, we used as a starting point the Navier-Stokes momentum conservation equations for a Newtonian, incompressible fluid without buoyancy effects (which is appropriate for the test case used in this study, see Sect. 3.1):

$$\frac{\partial u_j}{\partial t} = -\frac{\partial u_i u_j}{\partial x_i} - \frac{1}{\rho_0}\frac{\partial P}{\partial x_j} + \nu\frac{\partial^2 u_j}{\partial x_i^2}, \tag{1}$$

where $u_j$ $(u, v, w)$ [m s$^{-1}$] is the wind velocity along the j-th direction, $t$ [s] the time, $x_i$ and $x_j$ [m] the positions in the i-th direction and j-th direction respectively, $\rho_0$ [kg m$^{-3}$] the density, $P$ [Pa] the pressure, and $\nu$ [m$^2$ s$^{-1}$] the kinematic viscosity.

The governing LES equations are usually derived by applying a *generic*, unspecified filtering operation $\boldsymbol{G}$ to Eq. 1, which introduces a subgrid term $\tau_{ij} \equiv \overline{u_i u_j} - \overline{u_i}\ \overline{u_j}$ that has to be modeled (Pope, 2001; Sagaut, 2006). Traditional subgrid models like Smagorinsky (Smagorinsky, 1963; Lilly, 1967) attempt to model $\tau_{ij}$ associated with $\boldsymbol{G}$. However, by only considering the generic operation $\boldsymbol{G}$, they cannot directly compensate for the discretization errors arising on a specific finite numerical grid. Although the impact of the discretization errors can be reduced by adopting an explicit filtering technique (for instance by increasing the grid resolution compared to the filter width), this is in practice often not done because of the high computational

cost (Sagaut, 2006). It may therefore be beneficial to develop a LES SGS model that directly compensates for the introduced
discretization errors, ideally such that explicit filtering is not required anymore.

To this end we applied the *finite volume* filtering operation $\boldsymbol{G^{FV}}$ to Eq. 1 instead of the *generic* operation $\boldsymbol{G}$. $\boldsymbol{G^{FV}}$ is
defined as a 3D top-hat filter sampled on an a priori defined finite volume grid, where the finite sampling implicitly imposes
an additional spectral cutoff filter (Langford and Moser, 1999; Zandonade et al., 2004). We used $\boldsymbol{G^{FV}}$ to derive an alternative
set of LES equations (Eq. 3) that reflects the employed finite volume grid, and removes the need for commutation between the
filtering and spatial differentiation operators (Eq. 3; Denaro (2011)). This allowed us to explicitly include many instantaneous
discretization errors in the definition $\tau_{ij}$, making use of prior knowledge about the employed finite volume grid and numerical
schemes.

Considering for the sake of clarity only equidistant LES grids, following Zandonade et al. (2004) the filtered velocity
associated with $\boldsymbol{G^{FV}}$, $\overline{u_j}$, at a certain grid cell with indices $(\mathrm{l,m,n})$ can be written as:

$$\overline{u_j}(\mathrm{l,m,n}) = \frac{1}{\Delta x \Delta y \Delta z} \int\limits_{\Omega_j(\mathrm{l,m,n})} u_j(x,y,z) \, \mathrm{d}\boldsymbol{x'}, \tag{2}$$

where $\Delta x, \Delta y, \Delta z$ are the equidistant filter widths in the three spatial directions, $\Omega_j(\mathrm{l,m,n})$ the grid cube/control volume for
$u_j$ at the considered grid cell, and $\boldsymbol{x}$ a vector indicating the position $(x,y,z)$ in the flow domain. Since we focused in this
study on *staggered* finite-volume grids (Table 1), the location of each control volume $\Omega_j(\mathrm{l,m,n})$ depends on the $j$-component
considered (Sect. 3.2).

Applying the finite volume filter to Eq. 1, using the divergence theorem to convert the volume integrals to surface integrals,
and combining the advection and viscous stress terms, for a certain grid cell we get an expression similar to that obtained by
Zandonade et al. (2004):

$$\frac{\partial \overline{u_j}(\mathrm{l,m,n})}{\partial t} = -\frac{1}{\Delta x \Delta y \Delta z} \int\limits_{\partial \Omega_j(\mathrm{l,m,n})} (u_i u_j - \nu \frac{\partial u_j}{\partial x_i}) n_i \, \mathrm{d}\boldsymbol{x'} - \frac{1}{\Delta x \Delta y \Delta z} \frac{1}{\rho_0} \int\limits_{\partial \Omega_j(\mathrm{l,m,n})} p n_j \, \mathrm{d}\boldsymbol{x'}, \tag{3}$$

where $\partial \Omega_j(\mathrm{l,m,n})$ is the surface area of the control volume $\Omega_j(\mathrm{l,m,n})$, and $n_i$, $n_j$ the i-th and j-th component respectively of
the outward pointing normal vector $\boldsymbol{n}$ corresponding to $\partial \Omega_j(\mathrm{l,m,n})$. Noteworthy is that, by invoking the divergence theorem,
the divergence operator itself is effectively replaced by surface integrals, which removes the need for a commutative filter
(Denaro, 2011) and avoids the truncation errors introduced by the discretization of the divergence operator on the finite grid.

The well-known closure problem does, off course, persist. In fact, none of the terms on the right-hand side of Eq. 3 can be
determined exactly on the available finite LES grid and therefore have to be approximated. As argued however by Langford
and Moser (2001) and Zandonade et al. (2004), an *optimal* formulation for the pressure term is impractical and barely more
accurate than traditional finite-volume pressure schemes.

The errors made in approximating the time derivative, in turn, are usually constrained by the advection and diffusion terms
through the selected time step. Furthermore, the time discretization scheme we selected (Table 3.1) has good energy conserva-
tion properties with a slight damping of TKE over time (van Heerwaarden et al., 2017b).

In this study we will therefore only consider the instantaneous spatial discretization errors in the advection and viscous stress terms. We further note that, in contrast to eddy-viscosity SGS models, the isotropic part of the transport terms does not have to be incorporated in a modified pressure term.

To approximate the advection and viscous stress terms on the finite LES grid, in this study we used second-order linear interpolations (Sect. 3.2, 4.2). If we then consider specifically i) the control volume of the $u$-component, and ii) the transport in vertical direction, we can rewrite the first term at the right-hand side of Eq. 3 as follows:

$$
\frac{1}{\Delta x \Delta y \Delta z} \int_{\partial \Omega_u^{in}(l,m,n)} \left( wu - \nu \frac{\partial u}{\partial z} \right) \mathrm{d}x' \mathrm{d}y' - \frac{1}{\Delta x \Delta y \Delta z} \int_{\partial \Omega_u^{out}(l,m,n)} \left( wu - \nu \frac{\partial u}{\partial z} \right) \mathrm{d}x' \mathrm{d}y'
$$

$$
= \frac{1}{\Delta z} \left( \frac{\overline{w}(l,m,n) + \overline{w}(l-1,m,n)}{2} \frac{\overline{u}(l,m,n) + \overline{u}(l,m,n-1)}{2} - \nu \frac{\overline{u}(l,m,n) - \overline{u}(l,m,n-1)}{\Delta z} \right.
$$

$$
\left. - \frac{\overline{w}(l,m,n+1) + \overline{w}(l-1,m,n+1)}{2} \frac{\overline{u}(l,m,n) + \overline{u}(l,m,n+1)}{2} + \nu \frac{\overline{u}(l,m,n+1) - \overline{u}(l,m,n)}{\Delta z} \right)
$$

$$
+ \frac{1}{\Delta z} \left( \tau_{wu}^{in}(l,m,n) - \tau_{wu}^{out}(l,m,n) \right), \tag{4}
$$

where $\partial \Omega_u^{in}(l,m,n)$ and $\partial \Omega_u^{out}(l,m,n)$ are, respectively, the lower and upper boundaries of the control volume corresponding to the $u$-component $\Omega_u(l,m,n)$, representing two different subsets of the total control volume area $\partial \Omega_u(l,m,n)$.

$\tau_{wu}^{in}(l,m,n)$ and $\tau_{wu}^{out}(l,m,n)$, in turn, are unknown terms at the lower and upper boundary of the considered control volume, and are directly defined by the given Eq. 4. They correct for the unresolved physics and instantaneous discretization errors introduced by the employed approximations denoted at the right-hand side.

The correction terms for the other control volumes and transport components can be defined in a similar manner. In the remainder of the paper, we will denote the complete correction terms with the shorthand notation $\tau_{ij}^{in}$ and $\tau_{ij}^{out}$. It is these complete terms we aim to predict with our ANN-based LES SGS model. To fully solve Eq. 3, *after* training our ANN SGS model only makes use of information available in actual finite-volume LES: for its inputs, it relies only on the resolved flow fields $\overline{u}, \overline{v}, \overline{w}$ and their boundary conditions (Sect. 3.3).

## 3 Methodology

In this section, we will describe in detail the implementation of our ANN SGS model. First, we will provide a description of the DNS test case we used to train and test our ANN SGS model (Sect. 3.1). Next, we will briefly outline how we generated the data needed to train our ANN SGS model, using the selected DNS test case (Sect. 3.2). Subsequently, we will describe how we designed and trained our ANN SGS model (Sect 3.3 and Sect. 3.4). Finally, we will specify how we tested the a priori and a posteriori performance of our ANN SGS model (Sect. 3.5).

## 3.1 DNS test case

We used as a test case a DNS of incompressible neutral channel flow (with friction Reynolds number $Re_\tau$ being equal to 590) based on Moser et al. (1999). The friction Reynolds number is a variant of the standard Reynolds number based on the friction velocity, which is typically lower in magnitude than the standard Reynolds number and is often used in the context of wall-bounded turbulent flows (e.g. Pope, 2001). The friction velocity, in turn, is a velocity scale that measures the amount of mechanically-generated turbulence, and consequently is a logical scale to consider in neutral channel flow. We note that the selected friction Reynolds number is relatively low compared to most turbulent flows occurring in nature.

As simulation tool we used the high-order DNS and finite-volume LES code MicroHH (v2.0), which has been verified previously for the case selected in this study (van Heerwaarden et al., 2017b). The selected neutral channel flow is a turbulent flow bounded by walls at both the bottom and top of the domain (no-slip boundary conditions), with a mean flow characterized by a symmetric horizontally averaged vertical profile (Fig. (1)). In the horizontal directions, periodic boundary conditions were applied and a constant volume-averaged velocity ($U_f = 0.11\mathrm{m\ s^{-1}}$) was enforced by dynamically adjusting the pressure gradient.

We stored in total 31 3D flow fields of the wind velocity fields $u, v, w$ at time intervals of 60s after the flow reached steady state. This time interval was large enough to ensure that subsequent stored flow fields were (nearly) independent, which is preferable for the training and testing of the neural networks (Sect. 3.4 and 3.5). More details about the used simulation set-up and simulation code can be found in Table 1 and van Heerwaarden et al. (2017b).

**Table 1.** Simulation specifications for direct numerical simulation of incompressible neutral channel flow test case we used to generate the training data (Sect. 3.2). Here, $\delta$ [m] refers to the channel-half width. Additional details about the employed code (MicroHH v2.0) are given in van Heerwaarden et al. (2017b).

| | |
|---|---|
| Friction Reynolds number $Re_\tau$ | 590 |
| Boundary conditions | horizontal directions (x,y): periodic, vertical direction (z): no-slip |
| Domain size(x,y,z) | $2\pi\delta,\ \pi\delta,\ 2\delta$ |
| Kinematic viscosity $\nu$ | $1.0 \times 10^{-5}\ [\mathrm{m^2\ s^{-1}}]$ |
| Prescribed volume-averaged velocity $U_f$ | $0.11\ [\mathrm{m\ s^{-1}}]$ |
| Grid resolution (x,y,z) | $768,\ 384,\ 256$ (stretched in vertical) |
| Employed grid | staggered Arakawa C-grid |
| Spatial discretization | fourth-order interpolation scheme |
| Time discretization | three-stage, third-order Runge-Kutta scheme |

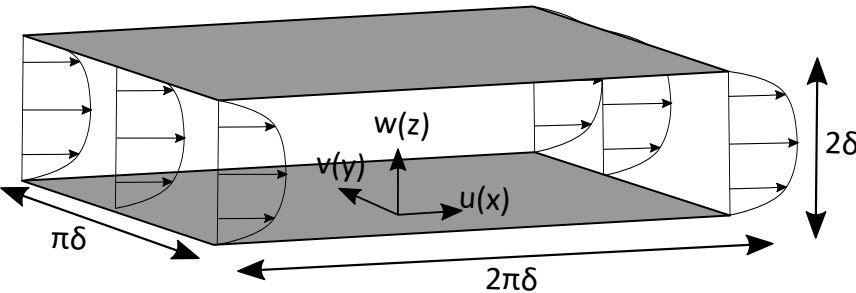

**Figure 1.** Sketch of simulated turbulent channel flow. Here, $\delta$ [m] refers to the channel-half width.

## 3.2 DNS training data generation

Using the filtering procedure outlined in Sect. 2, we calculated consistent pairs of i) low-resolution flow velocity fields $\overline{u_j}$ (that serve as input for the ANN), and ii) correction terms $\tau_{ij}^{in}$, $\tau_{ij}^{out}$ (that serve as the ground truth for the ANN predictions) from 31 previously stored DNS flow fields (Sect. 3.1). We used these input-output pairs as training data for our ANNs (Sect. 3.4).

By design, the filter in Sect. 2 is directly defined by a selected coarse equidistant LES resolution (Eq. 2). To generate the training data, we selected three different typical horizontal equidistant coarse resolutions with an identical coarsening in the vertical: $192 \times 96 \times 64$, $96 \times 48 \times 64$, and $64 \times 32 \times 64$ (x × y × z) cells. These three resolutions correspond to horizontal coarse-graining factors, $f_{hor}$, of 4, 8, and 12 respectively. In the remainder of the paper, we will denote the horizontal coarse-graining factor(s) used during training and testing as $f_{hor,train}$ and $f_{hor,test}$ respectively (see Sect. 3.5.1).

We note that the spatial discretization errors introduced by the applied coarsening, specifically concern errors associated with typically applied second-order linear interpolations (Eq. 4). These interpolation errors remove a substantial fraction of the turbulent energy remaining after applying the filter (Eq. 2), reflecting their detrimental impact on the smallest resolved scales (Fig. 2). Only by including their impact in the predicted correction terms, our ANN SGS model is able to fully compensate for the spatial discretization errors in the advection and viscous stress terms.

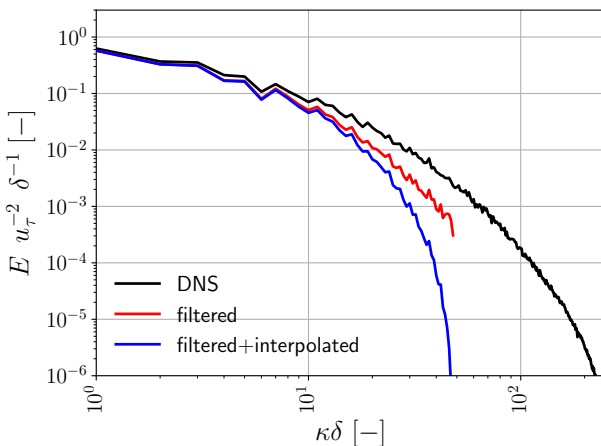

**Figure 2.** Example streamwise power spectra of $u$ for the selected channel flow (Sect. 3.1), at a height of $0.109\delta$ (i.e. in the log-layer) and considering a typical coarse equidistant LES resolution of $64 \times 32 \times 64$ (x × y × z) cells (which corresponds to a horizontal coarse-graining factor of $f_{hor} = 8$. Here, $\delta$ [m] refers to the channel-half width. The power spectral density $E$ on the vertical axis has been normalized by $\delta^{-1}$ and $u_\tau^{-2}$, where $u_\tau$ [m s$^{-1}$] is the friction velocity. Here, the black line corresponds to the power spectrum of the DNS fields, the red line to the power spectrum remaining after the finite-volume LES filter (Eq. 2) has been applied, and the blue line the spectrum remaining after both the finite-volume filter and the second-order linear interpolations required on the coarse LES grid (Eq. 4).

### 3.3 ANN architecture

We used feed-forward, fully-connected ANNs with a single hidden layer, to predict the correction terms $\tau_{ij}^{in}$ and $\tau_{ij}^{out}$ with the resolved flow fields $u_j$ as input. These are simple ANNs that facilitate computationally fast evaluations and easy implementation. We did not use deeper, more sophisticated ANNs to limit the computational cost involved in making predictions with the ANN as much as possible. This computational cost is critical for the affordability of an ANN SGS model in an actual LES simulation (Sect. 4.2).

To introduce non-linearity in the ANN, we used as an activation function the leaky rectified linear unit (ReLu) function (Maas et al., 2013) with the constant $\alpha$ set to the common value $0.2$. This non-linear activation function, together with the linear matrix-vector multiplications and bias parameter addition, defines the entire functional form of the ANN.

Similar to conventional LES SGS models, the ANN should preferably act on a small subdomain of the full grid to facilitate integration in our simulation code MicroHH, which uses domain decomposition for distributed memory computing. We consequently predicted with the ANN only the $\tau_{ij}^{in/out}$-values associated with one grid cell $(l, m, n)$ at a time. As input to the ANN, we used the locally resolved flow fields $\overline{u_j}$ in a $5 \times 5 \times 5$ stencil surrounding the grid cell for which we predict $\tau_{ij}^{in/out}$. Similar to Cheng et al. (2019) and Yang et al. (2019), we opted not to make our inputs Galilean/rotational invariant as the walls already provide an intrinsic coordinate system and velocity reference.

To select appropriate $5 \times 5 \times 5$ inputs stencils close to the boundaries of the domain, we made use of the horizontal periodic boundary conditions and the vertical no-slip conditions. We encoded the no-slip conditions in the input stencils by mirroring

$\overline{u_j}$ over the walls, such that $\overline{u_j}$ linearly interpolated to the wall was $0$ m s$^{-1}$. This may have helped the ANN to distinguish the near-wall region from the bulk of the flow, potentially removing the need for separate SGS and wall models.

Using the $5{\times}5{\times}5$ stencils in combination with the employed staggered Arakawa C-grid, an asymmetric bias is introduced in the ANN input and output variables if no special care is taken. We overcame this issue by combining three separate single-layer ANNs, where each one corresponded to one of the three control volumes considered (Sect. 2). Here, each received a stencil with slightly adjusted dimensions, and predicted only the correction terms ($\tau_{ij}^{in}, \tau_{ij}^{out}$) corresponding to the considered control volume (resulting in 6 outputs per ANN; Fig. 3). This ensured symmetry in the inputs and outputs of the ANN (Fig. 4 panel (a)), and did not increase the computational effort involved in evaluating the ANN after training.

In fact, this allowed us to reduce the number of ANN evaluations in the a posteriori simulation (Sect. 4.2) by almost a factor 2. Except for close to the walls, evaluating the ANN with a checkerboard-like pattern was sufficient to obtain all the needed correction terms (Fig. 4 panel (b)). Close to the walls, we did require (sometimes partial) ANN evaluations at every grid cell to calculate all needed correction terms: the checkerboard-like pattern does not provide all the correction terms at the edges of the domain. In the horizontal directions, we could make use of the periodic boundary conditions at the edges of the domain.

**Figure 3.** Architecture of ANN-framework used in this study. We combined three separate ANNs that each correspond to one of the three considered control volumes. For more information, please refer to Sect. 3.3.

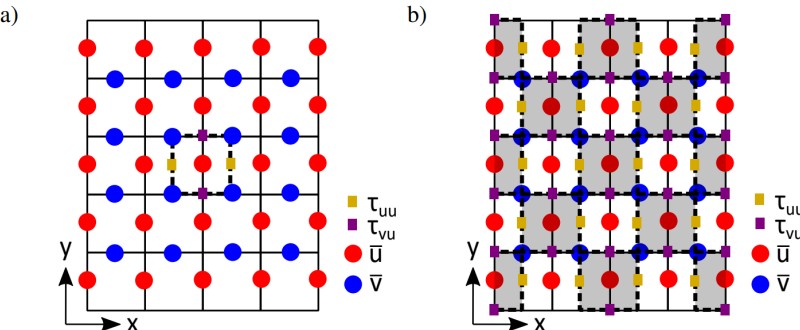

**Figure 4.** Panel a): Example two-dimensional input stencil of $\overline{u}, \overline{v}$ that the ANN corresponding to the control volume of $u$ receives, together with four of its outputs (i.e. $\tau_{uu}^{in}$, $\tau_{uu}^{out}$, $\tau_{uv}^{in}$, $\tau_{uv}^{out}$). Panel b): Two-dimensional visualization of the way we evaluated the ANN during a posteriori simulations. By evaluating the ANN in checkerboard-like pattern (i.e. only evaluating the grey-shaded grid cells) and making use of the periodic boundary conditions, we could calculate all needed correction terms except those close to the walls.

### 3.4 ANN training

We trained the employed ANNs (Fig. 3) using the training data (consisting of corresponding local $5 \times 5 \times 5$ $\overline{u_j}$ fields and correction terms $\tau_{ij}^{in/out}$; Sect. 3.2) we generated from 31 previously stored DNS flow fields (Sect. 3.1). The exact number of unique samples we could extract from each flow field during training, depended on the considered $f_{hor,train}$ (Sect. 3.2). For the case we mostly focused on in the a priori and a posteriori test (i.e. where $f_{hor,train} = 8$; see Sect. 3.5), we could extract 294.912 unique samples from each flow field. Of the 31 stored flow fields, we used 25 for training, 3 for validation during training and tuning of the the hyperparameters, and 3 for the a priori and a posteriori test (Sect. 3.5.1).

To train our ANNs, we used TensorFlow (v 1.12.0), an open-source machine learning framework (Abadi et al., 2016). We relied on the backpropagation algorithm (Rumelhart et al., 1986) incorporated within TensorFlow to minimize the loss function. We defined the loss function as the mean squared error (MSE) between the 18 DNS-derived $\tau_{ij,DNS}^{in/out}$-components (Sect. 3.2), and the 18 ANN-predicted $\tau_{ij,ANN}^{in/out}$-components (Sect. 3.3), combining the results from all three separate ANNs (Sect. 3.3). We observed good convergence of both the training and validation loss without signs of over-fitting for all the ANNs we tested (shown as an example for $f_{hor,train} = 8$ in Fig.5).

Here, we chose the popular ADAM optimizer (Kingma and Ba, 2014) with a relatively low value for the learning rate $\eta$ (0.0001) and a relatively large batch size of 1000. As our training data contains a high amount of noise inherent to turbulence, these parameter choices were in our case needed to stabilize the training results and achieve good convergence. For all the chosen ANNs corresponding to 2 or 3 $f_{hor,train}$ (see Sect. 3.5.1), we ensured that the samples originating from the different $f_{hor,train}$ were approximately equally represented in each training batch.

Besides that, we implemented preferential sampling near the walls: during training, we selected the five horizontal layers closest to the bottom and top wall more often than the other horizontal layers (starting from the bottom/top wall towards the center of the channel, respectively with a factor 10, 8, 6, 4, and 2). The preferential sampling restored the balance in the training

data set between the physics near the wall and the bulk of the flow, allowing the ANN to improve its performance close to the walls where a SGS model generally matters most.

In Table 2 we give an overview of all the hyperparameters and settings we used. The chosen initialization methods for the weights and bias parameter are standard for the architecture and activation function we selected. Furthermore, in line with common practice, we normalized all the inputs and outputs with their means and standard deviations. This improved the convergence during training and accelerated learning.

**Table 2.** Fixed hyperparameters and settings used in the ANNs we trained. Here, # means 'number of'.

| | |
|---|---|
| # training iterations (epochs) | 500.000 ($\approx 38$ epochs for $f_{hor,train} = 8$, taking into account the preferential sampling) |
| # hidden layers | 1 |
| Batch size | 1000 |
| Loss function | mean squared error, no regularization |
| Activation function | Leaky ReLu with $\alpha = 0.2$ (Maas et al., 2013) |
| Optimizer | ADAM with $\beta_1 = 0.9$, $\beta_2 = 0.999$, and $\epsilon = 1e - 08$ (Kingma and Ba, 2014) |
| Learning rate $\eta$ | 0.0001 |
| Normalization | z-score ($\frac{\text{value} - \text{mean}}{\text{standard deviation}}$) |
| Weight/kernel initializer | He uniform variance scaling initializer (He et al., 2015) |
| Bias initializer | zeros initializer |

We performed a more extensive sensitivity analysis with the number of neurons in the hidden layer, $n_{hidden}$, as it is for our architecture a good measure of the model complexity. In general we found for all three selected $f_{hor}$ (Sect. 3.2) that increasing $n_{hidden}$, and thus increasing the model complexity, improved the reduction of the loss function without showing signs of over-fitting (shown as an example for $f_{hor} = 8$ in Fig. 5). However, the improvement in training loss reduction clearly reduced with increasing model complexity, while a higher model complexity increases the computational cost of the ANN SGS model. In the next sections we will therefore focus on the results we obtained with $n_{hidden} = 64$, as a reasonable compromise between accuracy and total computational cost.

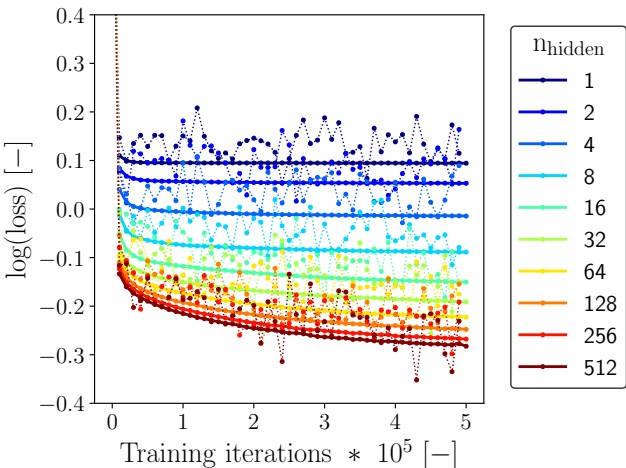

**Figure 5.** Evolution of the loss corresponding to the considered training batches (dotted lines) and the 3 validation flow fields (solid lines) with a changing number of neurons in the single hidden layer as a function of training iteration, for the ANNs using $f_{hor,train} = 8$ (see Sect. 3.5.1). To improve readability and keep the total computational effort involved in the training feasible, we show here both losses only for every 10.000 iterations instead of every single iteration.

## 3.5 ANN testing

### 3.5.1 A priori (offline) test

To assess the potential a priori accuracy of our ANN SGS model, we first compared the ANN predictions to the DNS-derived values (Sect. 2 and Sect. 3.2) for 3 flow fields held out during training (Sect. 3.4) and a single representative coarse LES

resolution (i.e. an equidistant grid with $f_{hor,train} = f_{hor,test} = 8$; see Sect. 4.1.1). This tests the ability of our ANN SGS model to generalize towards previously unseen realizations of the steady state associated with the selected channel flow (Sect. 3.1).

     We especially focused, in the log-layer, on $\tau_{wu}$ and the net energy transfer towards the unresolved scales, $\epsilon_{SGS}$, where $\epsilon_{SGS}$ is defined and approximated as $\epsilon_{SGS} \equiv -\tau_{ij}\overline{S_{ij}} \approx -\tau_{ij}^{in/out,int}\frac{\Delta \overline{u_j}}{\Delta x_j}$. We calculated $\epsilon_{SGS}$ by interpolating all the individual

components to the grid centers (denoted here as $\tau_{ij}^{in/out,int}$), and subsequently summing them. $\epsilon_{SGS}$ can be both positive and negative, where positive values indicate SGS dissipation and negative values back-scatter towards the resolved scales. Both these processes are critical for the a priori and a posteriori performance: dissipating sufficient energy to the unresolved scales is crucial for achieving stable a posteriori results. $\tau_{wu}$, in turn, is also of particular interest in channel flow: it is the vertical gradient of $\tau_{wu}$ that has to balance the imposed horizontal pressure gradient (e.g. Pope, 2001), making $\tau_{wu}$ critical

for the quality of the achieved steady state solution. The log-layer is mainly interesting because of its universal character. In the log-layer, the horizontally averaged profiles of the mean velocity and Reynolds stress tensor components becomes partly independent of the Reynolds number when properly scaled with wall units (e.g. Pope, 2001).

As a reference, we included in the comparison the subgrid fluxes and net SGS transfer predicted with the popular Smagorinsky (Lilly, 1967) SGS model (see Sect. 4.1.2), which we will denote as $\tau_{ij,Smag}$ and $\epsilon_{smag}$ respectively. In the Smagorinsky SGS model, $\tau_{ij,Smag}$ is modelled as $\tau_{ij} = -2\nu_r \overline{S_{ij}}$, with $\nu_r$ being the modelled eddy-viscosity coefficient and $\overline{S_{ij}}$ being the filtered strain rate tensor (defined as $\overline{S_{ij}} \equiv \frac{1}{2}\left(\frac{\partial \overline{u_i}}{\partial x_j} + \frac{\partial \overline{u_j}}{\partial x_i}\right)$) (Pope, 2001, e.g.). In line with usual practice for wall-bounded flows, we augmented the model for $\nu_r$ with an ad-hoc Van Driest (Van Driest, 1956) wall-damping function to (partly) compensate for the known over-dissipative behaviour close to walls (e.g. Pope, 2001; Sagaut, 2006). Consequently, $\nu_r$ is effectively modelled as $\nu_r = \left(c_s \Delta \left(1 - \exp\left(-z^+/A^+\right)\right)\right)^2 \overline{S}$, with $c_s$ being the Smagorinsky coefficient (which is being set to 0.1), $\Delta$ being the filter size (defined as $\Delta \equiv (\Delta x \Delta y \Delta z)^{\frac{1}{3}}$), $z^+$ the absolute vertical distance from the closest wall normalized by $u^*$ and $\delta$, $A^+$ an empirical constant (which is being set to 26), and $\overline{S}$ the squared filtered strain rate tensor (defined as $\overline{S} = 2\left(S_{ij}S_{ij}\right)^{\frac{1}{2}}$).

To facilitate easier interpretation and comparison with the Smagorinsky SGS model, for the ANN and DNS results we combined the two separate correction terms $\tau_{ij}^{in}, \tau_{ij}^{out}$. In the remainder of the paper we will denote the resulting combined correction terms as $\tau_{ij,ANN}$ and $\tau_{ij,DNS}$, where both consist of the same 9 components as $\tau_{ij,smag}$. We did this in accordance with the way we evaluated the ANNs within our CFD-code MicroHH during the a posteriori test (Sect. 3.3).

On top of the comparison for a single horizontal coarse resolution, we separately explored the generalization performance of the developed ANN SGS model with respect to the selected coarse horizontal resolution in Sect. 4.1.3 . To this end, we trained our ANN SGS model, in three different ways, on filtered DNS data corresponding to all selected $f_{hor}$ (4, 8, 12 respectively; see Sect. 3.2):

1. Train only on filtered DNS data corresponding to $f_{hor} = 8$.

2. Train on filtered DNS data corresponding to $f_{hor} = 4, 12$.

3. Train on filtered DNS data corresponding to all three $f_{hor}$.

For all three training configurations mentioned above, we tested the performance of the ANN SGS models on previously unseen filtered data corresponding to all three $f_{hor}$. This thus includes several cases where the ANN SGS model is being applied to other resolutions than seen during training.

Finally, to get some more insight into the behaviour of our ANNs, in Sect. 4.1.4 we calculated for every input variable in the $5 \times 5 \times 5$ stencils the so-called permutation feature importance (e.g. Fisher et al., 2019; Molnar, 2019; Breiman, 2001) associated with predicting $\tau_{wu}^{in}$ and $\tau_{wu}^{out}$ in the log-layer.

The most important advantage of these permutation feature importances is their intuitive meaning: they indicate how important a certain input variable is for the prediction quality of the $\tau_{wu}^{in}, \tau_{wu}^{out}$ in the log-layer: the higher it is, the more important that variable is. Specifically, the permutation feature importance measures by which factor the prediction error (in our case measured as the root-mean square error between the DNS-values and ANN predictions) increases when the information contained in that input variable is destroyed, while the information in the other input variables is retained. We destroyed the information in each input variable by randomly shuffling it in the corresponding horizontal plane. Besides that, we averaged the calculated permutation feature importances over all the 3 testing flow fields and over 10 different random shufflings, to stabilize the results.

We intentionally chose not to shuffle the input variables along different heights. Because of the strong mean vertical gradient in $u$, this would possibly introduce an unrealistic bias into the calculated permutation feature importances. We do emphasize that the permutation feature importances are likely affected by the correlations existing in our input data. The permutation feature importances we report therefore need to interpreted with caution.

### 3.5.2 A posteriori (online) test

To test the a posteriori performance of our ANN LES SGS model, we directly incorporated one of our ANNs (i.e. with $n_{hidden} = 64$ and $f_{hor,train} = 8$) into our CFD code MicroHH (v2.0) (van Heerwaarden et al., 2017b). We chose the input and output variables of our ANN SGS model such that the integration into our CFD code was relatively straightforward (Sect. 3.3). Furthermore, we improved the computational performance of the ANN SGS model by implementing BLAS routines from the Intel(R) Math Kernel Library (version: 2019 update 5 for Linux), which has been optimized for the Intel CPUs we used (i.e. E5-2695 v2 (Ivy Bridge) and E5-2690 v3 (Haswell)). Still, the computational effort involved in the ANN SGS model was large: an equivalent LES simulation with the Smagorinsky SGS model was for our set-up about a factor 15 faster, showing that, in its current form, our ANN SGS model still needs more optimizations for practical applications.

With the ANN SGS model incorporated in our CFD code, we ran a LES with an equidistant grid of $96 \times 48 \times 64$ cells, directly corresponding to the selected $f_{hor,train} = 8$, for the turbulent channel flow test case described in Sect. 3.1. Here, we used second-order linear interpolation to calculate all the velocity tendencies, consistent with our filtering and training data generation procedure (Sect. 2 and 3.2). Furthermore, we initialized the LES simulation from one of the 3 flow fields reserved for the a priori testing. We did this to ensure that any possible errors in the initialization phase of the LES (i.e. before steady state is achieved) did not impact the solution. Still, our LES ran freely from the prescribed initialized steady-state fields, meaning that all the model and discretization errors made in calculating the channel flow steady state dynamics were included.

## 4 Results & Discussion

In this section, we will characterize the a priori and a posteriori performance of our ANN SGS model. We will first describe the a priori performance of our ANN SGS model and the Smagorinsky SGS model for a single coarse resolution (i.e. where $f_{hor,train} = f_{hor,test} = 8$; Sect. 4.1.1 and 4.1.2). Subsequently, we will discuss the generalization performance of our ANN SGS model with respect to the selected coarse resolution (Sect. 4.1.3), and the permutation feature importances associated with the input stencils (Sect. 4.1.4). Finally, we will describe and discuss the instability we observed a posteriori (Sect. 4.2).

### 4.1 A priori (offline) test

#### 4.1.1 Single horizontal resolution ANN performance

The ANN-predicted $\tau_{wu,ANN}$, $\epsilon_{SGS,ANN}$ (with $n_{hidden} = 64$, see Sect. 3.4) in the log-layer generally show excellent agreement with the DNS-derived values (Fig. 6 - 9). Especially the consistency we found in the horizontal cross-sections (Fig. 6,

7 panel a and b) is striking given the noisy spatial patterns of $\tau_{wu,DNS}$ and $\epsilon_{SGS,DNS}$, which the ANN reproduces quite accurately both qualitatively and quantitatively. Particularly noteworthy is its ability to accurately reproduce both negative and positive $\epsilon_{SGS,DNS}$, as these are associated with back-scatter and SGS dissipation respectively. These two processes are both critical for the quality of the a posteriori simulations (see Sect. 4.2).

We note that the found correspondence between correction and SGS transfer terms in the log-layer of neutral channel flow, is in agreement with the results of Cheng et al. (2019), Park and Choi (2021), and Gamahara and Hattori (2017), despite that our training data generation procedure additionally accounted for numerical errors associated with LES where a staggered finite-volume grid acts as an implicit filter (Sect. 2 and 3.2). Consistent with the matching horizontal cross-sections, the ANN reproduces quite well the distributions and spectra of $\tau_{wu,DNS}$ and $\epsilon_{SGS,DNS}$ (Fig. 8, 9 panel b and c). The notable high
normalized spectral density of $\tau_{wu,DNS}$ at high wave modes, is a direct consequence of the instantaneous spatial discretization errors we compensate for. As these discretization errors remove a large part of the variance at the smallest resolved spatial scales (Fig. 2), the corresponding correction terms, including $\tau_{wu}$, are characterized by strong variability at the smallest resolved scales.

   From the tails of the distribution and the high wave modes of the spectra (Fig. 8 and 9 panel b), it is apparent that the ANN
does still slightly underestimate the extremes at small spatial scales characteristic of $\tau_{wu,DNS}$ and $\epsilon_{SGS,DNS}$. Probably, these extremes were hard to predict accurately because of their high stochastic nature and inherent rare occurrence. Yang et al. (2019) identified this issue in the context of an ANN-based LES wall model, and found that this issue persisted even when the errors were weighted inversely proportional to their PDF (i.e. giving extreme values larger weights in the loss function).

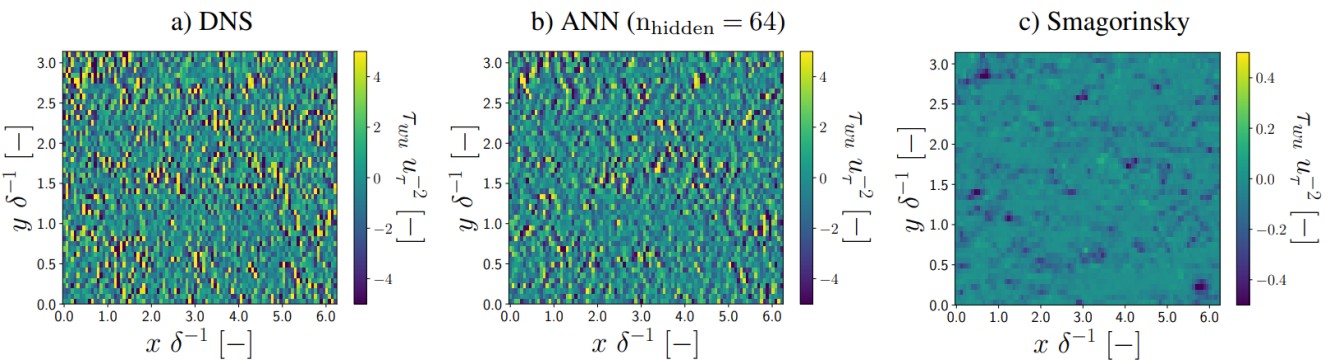

**Figure 6.** Horizontal cross-sections of $\tau_{wu}$ in the log-layer ($0.09375\frac{z}{\delta}$ ($55.3125z^+$)) for a representative flow field not used to train and validate the ANNs. All values are normalized by the friction velocity $u_\tau$ and half-channel width $\delta$.

.

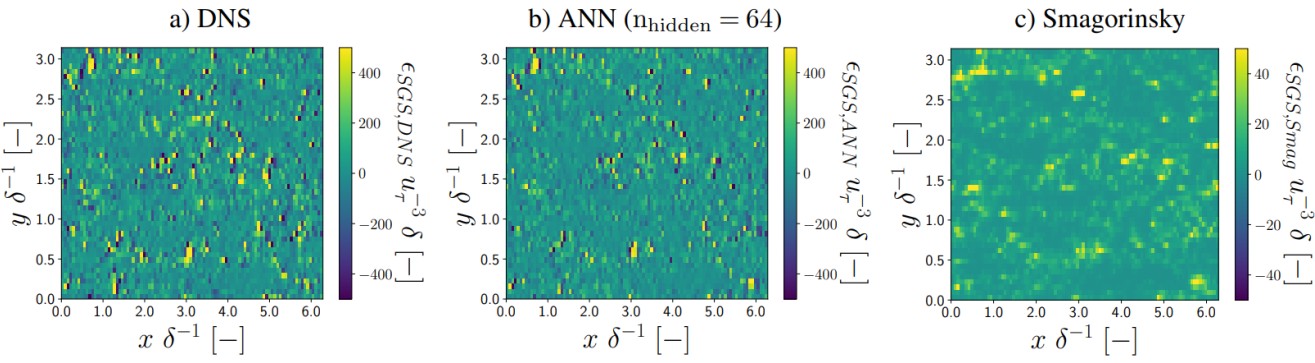

**Figure 7.** Horizontal cross-sections of $\epsilon_{SGS}$ in the log-layer $(0.109375\frac{z}{\delta}$ $(64.53125z^+))$ for a representative flow field not used to train and validate the ANNs. All values are normalized by the friction velocity $u_\tau$ and half-channel width $\delta$.

.

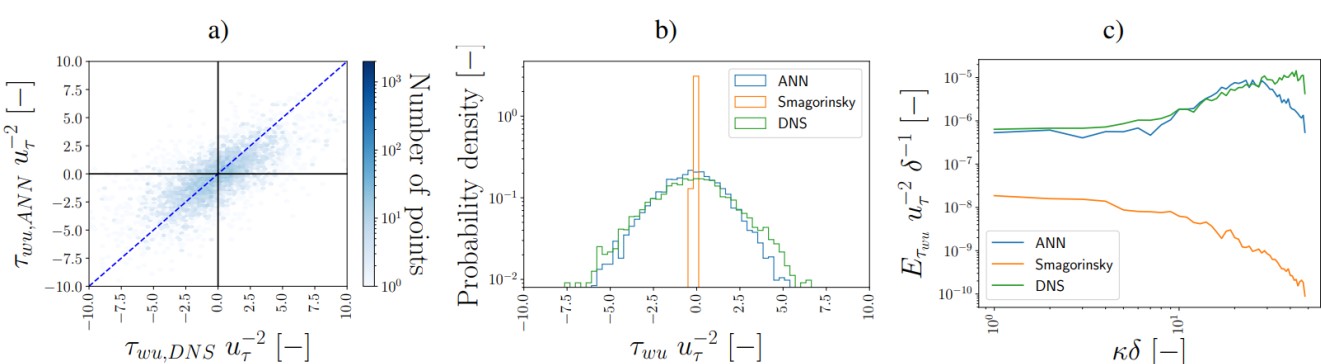

**Figure 8.** Performance of $\tau_{wu,ANN}$ (with $n_{hidden} = 64$) in the log-layer $(0.09375\frac{z}{\delta}$ $(55.3125z^+))$ for a representative flow field not used to train and validate the ANNs. Panel (a) shows the corresponding hexbin plot between $\tau_{wu,ANN}$ and $\tau_{wu,DNS}$, where the dotted blue line indicates the 1:1 line. Panel (b) shows the probability density functions, and panel (c) the streamwise spectra averaged in the spanwise direction. $\tau_{wu,ANN}$ and $\tau_{wu,DNS}$ have been normalized by the friction velocity $u_\tau^{-2}$. The power spectral density $E$ on the vertical axis in panel (c) has been normalized by $\delta^{-1}$ and $u_\tau^{-2}$. As a reference, in panel (b) and (c) $\tau_{wu,smag}$ is shown as well.

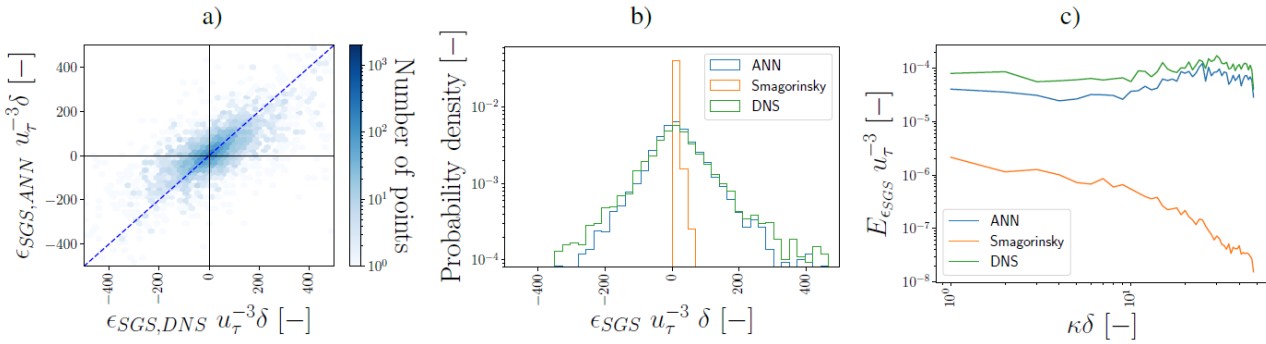

**Figure 9.** Performance of $\epsilon_{SGS,ANN}$ (with $n_{hidden} = 64$) in the log-layer ($0.109375\frac{z}{\delta}$ $(64.53125z^{+})$) for a representative flow field not used to train and validate the ANNs. Panel (a) shows the corresponding hexbin plot between $\epsilon_{SGS,ANN}$ and $\epsilon_{SGS,DNS}$, where the dotted blue line indicates the 1:1 line. Panel (b) shows the probability density functions, and panel (c) the streamwise spectra averaged in the spanwise direction. $\epsilon_{SGS,ANN}$ and $\epsilon_{SGS,DNS}$ have been normalized by the friction velocity $u_\tau^{-3}$ and $\delta$. The power spectral density $E$ on the vertical axis in panel (c) has been normalized by $u_\tau^{-3}$. As a reference, in panel (b) and (c) $\epsilon_{SGS,Smag}$ is shown as well.

Extending our focus from the log-layer to vertical profiles of horizontally averaged $\tau_{wu}$ and $\epsilon_{SGS}$, in general we again observe quite good correspondence between the ANN predictions and DNS-derived values (Fig. 10 and 11). In the profile of $\tau_{wu,ANN}$ we do see some deviations from the $\tau_{wu,DNS}$ profile, especially close to the walls. In our training data, the horizontally averaged flux of $\tau_{wu,DNS}$ was generally small compared to its point-wise fluctuations. As a result, the loss associated with $\tau_{wu,DNS}$ was probably more sensitive to the point-wise fluctuations than the average flux, which may have contributed to the observed deviations.

The vertical profile of $\epsilon_{SGS,ANN}$, in turn, matches very closely the profile of $\epsilon_{SGS,DNS}$. The ANN approximately provides the net dissipation inferred from the DNS, which primarily occurs close to the walls. Hence, this does not make yet clear why our ANN SGS model induces the observed a posteriori instability. In Sect. 4.2, we will elaborate more on potential reasons why our ANN SGS model nonetheless induces instability.

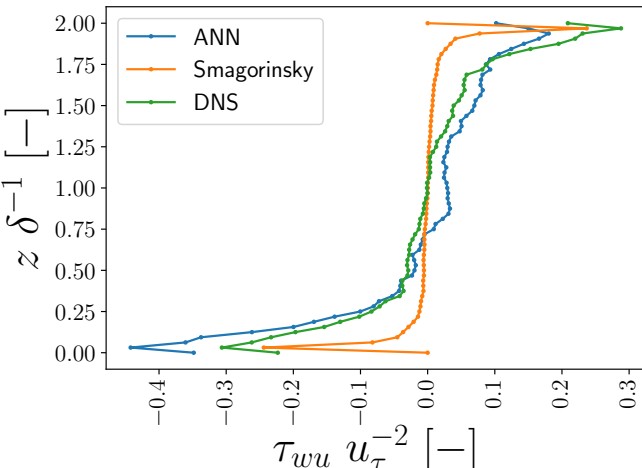

**Figure 10.** Vertical profiles of horizontally averaged $\tau_{wu,DNS}$ , $\tau_{wu,ANN}$, and $\tau_{wu,smag}$ at one representative time step not used to train and validate the ANNs. All values are normalized by the friction velocity $u_\tau^{-2}$ and half-channel width $\delta^{-1}$.

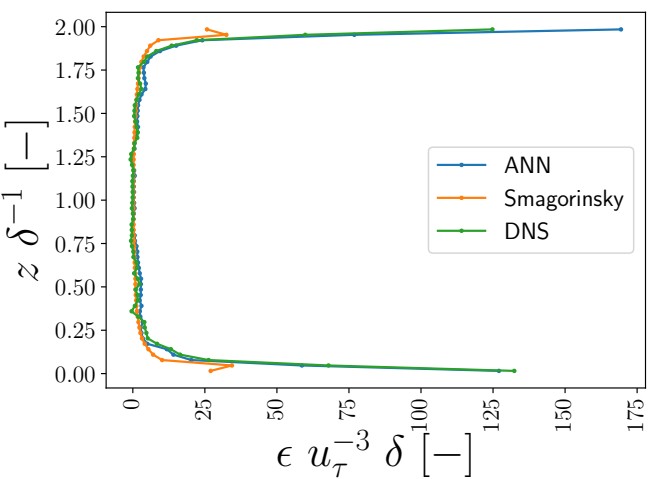

**Figure 11.** Vertical profiles of horizontally averaged $\epsilon_{SGS,DNS}$ , $\epsilon_{SGS,ANN}$, and $\epsilon_{SGS,smag}$ at one representative time step not used to train and validate the ANNs. All values are normalized by the friction velocity $u_\tau^{-3}$ and half-channel width $\delta$.

Extending our focus towards all components, we found that in general the ANN correlated well with all DNS-derived correction and SGS transfer terms (third row Table 3 and Fig. 12; mostly $\rho$ $0.6 - 0.9$). Looking more closely at the found correlations, we did find that the correlations differed depending on the channel height. Closer to the walls, the correlations generally slightly decreased compared to the middle of the channel (except for the vertical layers directly adjacent to the wall, where most terms show a better correlation). Here, we emphasize that we implemented a preferential sampling technique

(Sect. 3.4), which helped to minimize this reduction of prediction performance close to the walls compared to the middle of the channel.

Looking at the individual terms, some of them were clearly better predicted than others (e.g. $\tau_{vu}$ vs $\tau_{vw}$): this was likely related to differences in their magnitude that persisted even after the applied normalization (i.e. the same normalization was applied over the entire domain, meaning that some components with strong vertical gradients still contained more extreme values than components without a clear vertical gradient), and differences in their stochastic variability and consequent signal-to-noise ratio.

One clear outlier is $\tau_{wu}$ at the first vertical level (with $\rho = 0.339$, not shown), which appeared to be most difficult to predict. This component was located at the bottom wall because of the staggered grid orientation, and consequently only the viscous flux contributed. As a consequence, the target DNS values and input patterns were different than for other vertical levels and components, making it hard for the ANN to give accurate predictions. Still, the magnitude of the ANN predictions matched reasonably well the DNS values (not shown).

### 4.1.2 Single horizontal resolution Smagorinsky performance

Considering the individual grid points, the a priori performance of the Smagorinsky SGS model is in sharp contrast with the a priori ANN performance: $\tau_{ij,smag}$, and to a somewhat lesser extent $\epsilon_{SGS,smag}$, show barely any agreement with the DNS-derived values both qualitatively and quantitatively (Fig. 6 - 9).The poor point-wise a-priori performance of Smagorinsky is well-known in literature (e.g. Clark et al., 1979; McMillan and Ferziger, 1979; Liu et al., 1994). In addition, we can also observe its known inability, in the form we employed, to account for back-scatter (e.g. Pope, 2001; Sagaut, 2006).

In our case though, the point-wise a-priori performance of Smagorinsky is still worse than usually documented: the found correlations with DNS in our study (mostly $\rho = 0.0$ at individual heights for all correction and dissipation terms, not shown) are lower than reported before (where $\rho = \sim 0...0.4$; Cheng et al. (2019); Clark et al. (1979); McMillan and Ferziger (1979); Liu et al. (1994)). Furthermore, $\tau_{wu,Smag}$ and $\epsilon_{SGS,smag}$ are off by approximately one order of magnitude and are too smooth (Fig. 6, 7 and Fig. 8-9 panels b and c): in comparison to $\tau_{wu,DNS}$ and $\epsilon_{SGS,DNS}$, the PDF is narrower (Fig. 8,9 panel b), and the spectral energy in $\tau_{ij,Smag}$, $\epsilon_{SGS,smag}$ is smaller and skewed towards low wave modes (Fig. 8 and 9 panel c).

This exacerbated point-wise a priori performance of the Smagorinsky SGS model is caused by our alternative definition for $\tau_{ij}$, which, in contrast to the commonly defined $\tau_{ij}$, compensates for all the instantaneous discretization effects introduced by the staggered finite-volumes in both the advection and viscous flux terms (Sect. 2). As these discretization effects remove a large part of the variance present in the LES (Figure 2), our $\tau_{ij}$ inherently contains a large amount of variance that is not represented by Smagorinsky.

Focusing on the horizontally averaged vertical profiles of $\tau_{wu}$, we consequently found also that $\tau_{wu,smag}$ does not compare well with $\tau_{wu,DNS}$ (Fig. 10). Except close to the walls and the center of the channel, the Smagorinsky SGS model strongly underestimates the horizontally averaged $\tau_{wu}$. We emphasize that the correspondence close to the walls was only achieved because of the implemented ad-hoc Van Driest wall damping function (Van Driest, 1956).

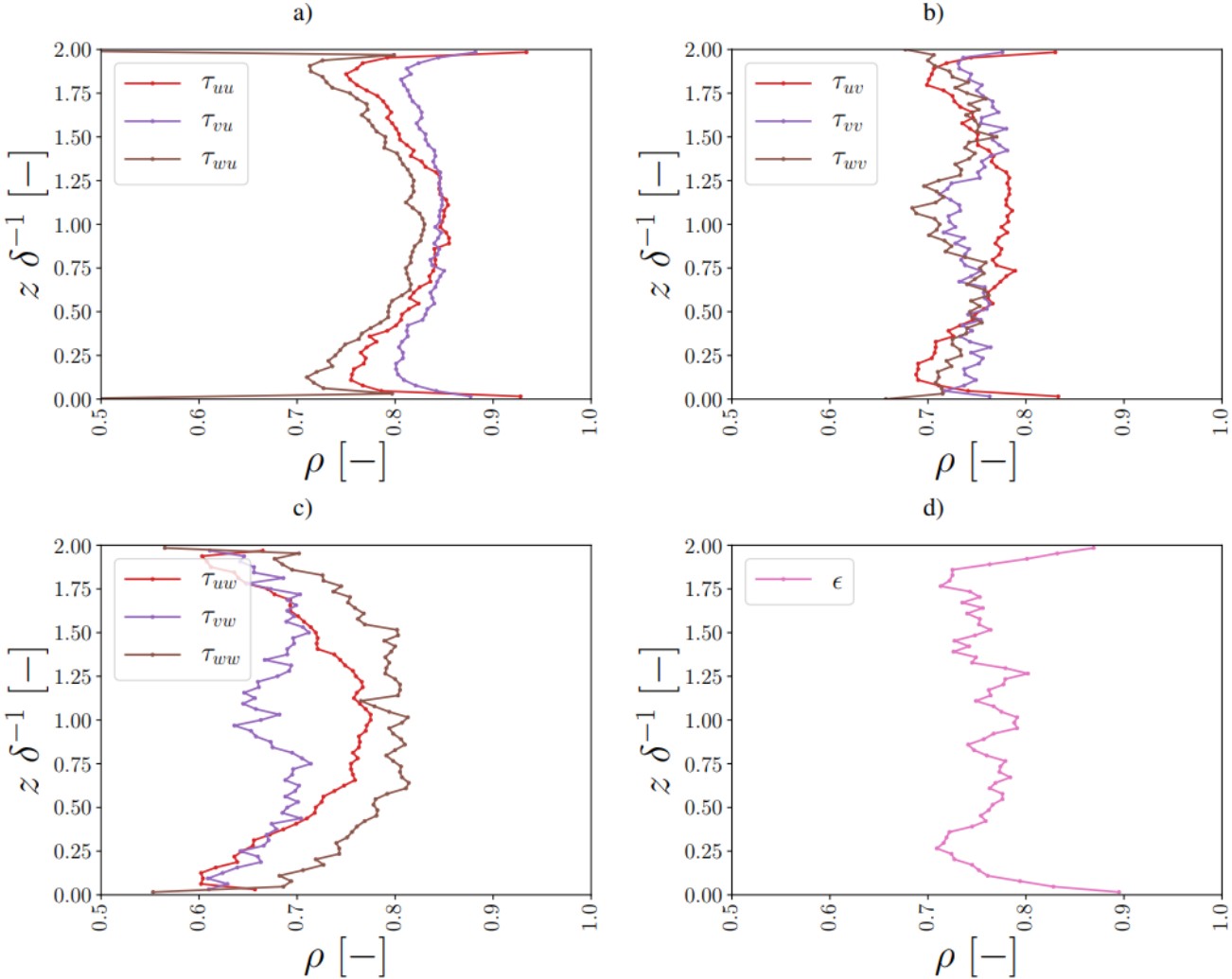

**Figure 12.** Vertical profiles of correlation coefficients between ANN predictions and DNS values for all correction and dissipation terms (panels a-d), at one representative time step not used to train and validate the ANNs. Here, the $j$-index refers to the considered control volume (Sect. 2). The heights are normalized by the half-channel width $\delta^{-1}$. Note that the $\tau_{uw,vw}$-components are left out at the first vertical level, as these are due to the staggered grid located exactly at the bottom wall. At the bottom-wall we imposed a no-slip boundary-condition, meaning that these components are by definition 0.

In the horizontally averaged vertical profiles of $\epsilon_{SGS}$ (Fig. 11), we observe a striking characteristic that may seem counter-intuitive at first: the Smagorinsky SGS model under-predicts $\epsilon_{SGS,DNS}$ at the walls, despite its known over-dissipative be-haviour in a posteriori tests (e.g. Pope, 2001; Sagaut, 2006). However, as the Smagorinsky SGS model does not directly com-

pensate for instantaneous discretization errors (and thus does not re-introduce the associated inherent variance), the Smagorin-sky SGS needs to produce less dissipation than our ANN SGS model to achieve stable a posteriori results (see also Sect. 4.2).

All in all, our ANN SGS model is clearly better able to represent $\tau_{ij,DNS}$ and $\epsilon_{SGS,DNS}$ in the presented a priori test than the Smagorinsky SGS model. This shows the promise ANN SGS models like ours could have to construct more accurate SGS

models that, in contrast to traditional SGS models like Smagorsinky, additionally compensate for instantaneous spatial dis-cretization errors. The most important issue remaining, is whether and how this a priori potential can be successfully leveraged in a posteriori simulations without introducing numeric instability.

### 4.1.3   Multiple horizontal resolutions ANN generalization

Overall, our ANN SGS model shows promising generalization capabilities towards other coarse horizontal resolutions than

the one considered in the previous section. The extent to which it is able to maintain its high a priori accuracy, however, does strongly depend on the considered $f_{train,hor}$ and $f_{test,hor}$ (Table 3).

**Table 3.** Pearson correlation coefficients $\rho$ between ANN predictions ($n_{hidden} = 64$) and DNS values averaged for all 3 test flow fields, all heights, and all correction and dissipation terms. The different coarse-graining factors used during training, $f_{hor,train}$, and testing, $f_{hor,test}$ are indicated. Here, the third row refers to the training and test configuration used in Sect. 4.1.1. More details about all the indicated training and test configurations, are given in Sect. 3.5.1.

| $f_{hor,train}$ | $f_{hor,test}$ | $\rho$ |
|---|---|---|
| 8 | 12 | 0.624 |
| 8 | 8 | 0.758 |
| 8 | 4 | 0.526 |
| 12, 4 | 12 | 0.656 |
| 12, 4 | 8 | 0.737 |
| 12, 4 | 4 | 0.832 |
| 12, 8, 4 | 12 | 0.657 |
| 12, 8, 4 | 8 | 0.744 |
| 12, 8, 4 | 4 | 0.832 |

Considering first the ANNs solely trained on $f_{hor,train} = 8$ (second-fourth row Table 3), we find, unsurprisingly, that they achieve their best performance when $f_{hor,test} = 8$ (which is identical to the configuration used in Sect. 4.1.1). More interest-

ingly however, we observe that these ANNs already have some generalization capability, even without having seen multiple

$f_{hor,train}$. This does depend on the selected $f_{hor,test}$: the performance is better for $f_{hor,test} = 12$ than for $f_{hor,test} = 4$ (where for multiple terms $\rho < 0.5$; not shown).

Comparing these ANNs to the ones trained on $f_{hor,train} = 4, 12$ (fifth-seventh row Table 3), first of all we see a clear, unsurprising improvement when $f_{hor,test} = 4, 12$: including the tested coarse-graining factors in the training, improves the ANN performance in the associated a priori test. Interestingly, this improvement is much larger for $f_{hor,test} = 4$ than for $f_{hor,test} = 12$.

Secondly, we observe that, without using $f_{hor,train} = 8$, the ANN performance on $f_{hor,test} = 8$ is only subtly lower than the ANNs directly trained on $f_{hor,train} = 8$. This shows that our ANN SGS model may accurately generalize to other unseen resolutions without losing its high a priori accuracy, if $f_{hor,test}$ is within the range of the used $f_{hor,train}$.

Finally comparing the previously discussed ANNs (with $f_{hor,train} = 4, 12$) to the ones trained on all three considered $f_{hor,train}$ (eight-tenth row Table 3), we find that additionally including $f_{hor,train} = 8$ barely influences the ANN performance for all terms (even when $f_{hor,test} = 8$). This again highlights the possibility our ANN SGS model may accurately generalize to other resolutions, as long as the range in the training data covers the testing situations. In doing so, the need to include multiple intermediate $f_{hor,train}$ can likely be limited.

### 4.1.4 Permutation feature importance ANN

For $\tau_{wu}^{in}$ and $\tau_{wu}^{out}$ in the log-layer, we calculated all the permutation feature importances associated with the ANNs listed in Table 3 (see Sect. 3.5.1). Generally, we found that highest feature importances were associated with $u$, and that the feature importances corresponding to ANNs trained on two or more resolutions were mostly lower than the ones corresponding to the ANNs trained on one resolution. The former suggests that the ANN focuses mostly on the flow velocity component in the mean direction, while the latter suggests that the ANN becomes less sensitive on the inputs when trained on multiple resolutions. As an example, we show in Fig. 13 and 14 the feature importances corresponding to the $u$-input stencil and the ANN only trained and tested on $f_{hor} = 8$ (third row Table 3).

Interestingly, all the calculated feature importances (including the ones not shown in Fig. 13 and 14) suggest that the input variables most important to the ANN are generally located close to the considered correction term. In addition, there seem to be an orientation along the mean flow direction $l$, with corresponding low feature importances at the edges in the span-wise $m$ and wall-normal $k$ direction.

Comparing the calculated feature importances corresponding to $\tau_{wu}^{in}$ and $\tau_{wu}^{out}$ in turn, we generally observe a corresponding shift in the vertical. For the shown $u$-velocity input stencil (Fig. 13-14) for instance, the vertical patterns corresponding to $\tau_{wu}^{in}$ and $\tau_{wu}^{out}$ are nearly mirrored versions of each other.

All in all, these findings suggest that the employed input stencils can be made smaller in the vertical and span-wise direction without sacrificing their predictive value, and that an extension along the stream-wise direction, in turn, may help to increase the predictive value of the input stencils.

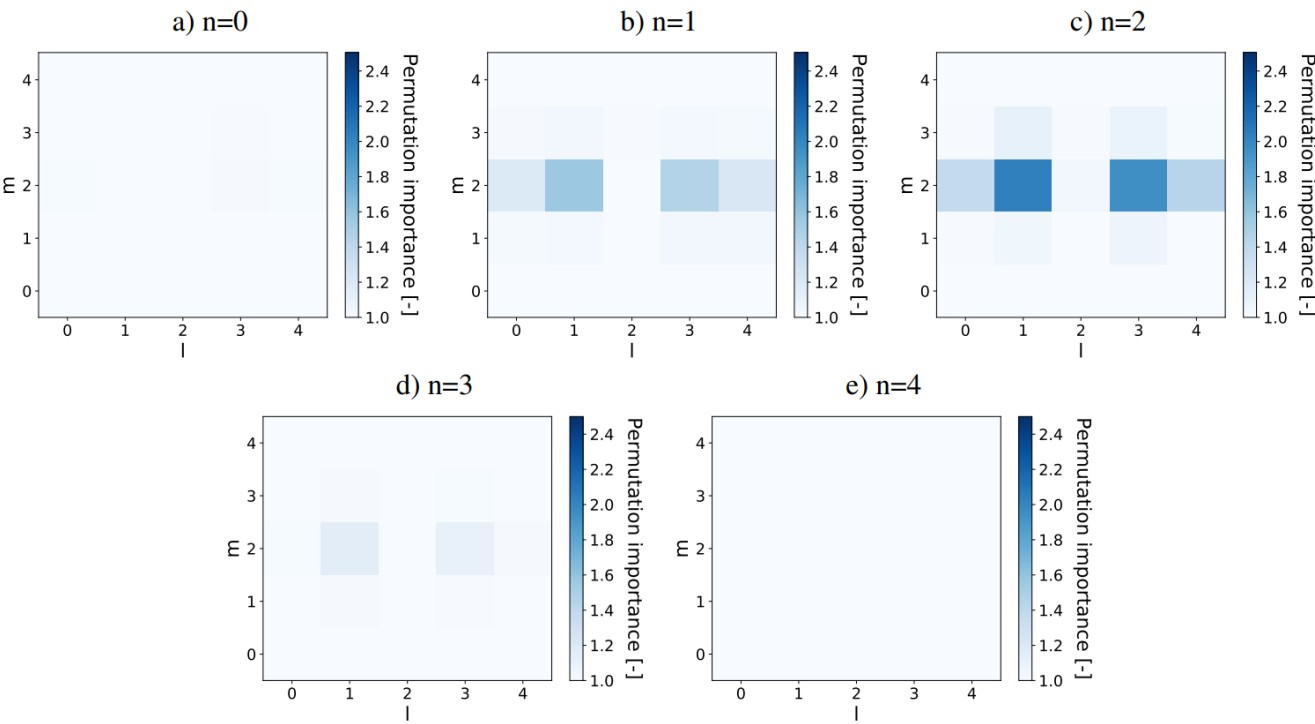

**Figure 13.** Permutation feature importance of all $u$-velocities in the local input stencil (with indices $(l, m, n)$) associated with predicting $\tau_{wu}^{in}$ in the log-layer ($0.09375\frac{z}{\delta}$ ($55.3125z^+$)) using an ANN (with $n_{hidden} = 64$) trained and tested only on $f_{hor} = 8$, averaged over 3 flow fields reserved for a priori testing and 10 random shufflings. The five panels a-e each show one of the five horizontal planes (indicated by their vertical index $n$) present in the input stencils. $\tau_{wu}^{in}$ is located in the center of the shown horizontal plane, halfway between $n = 1$ and $n = 2$.

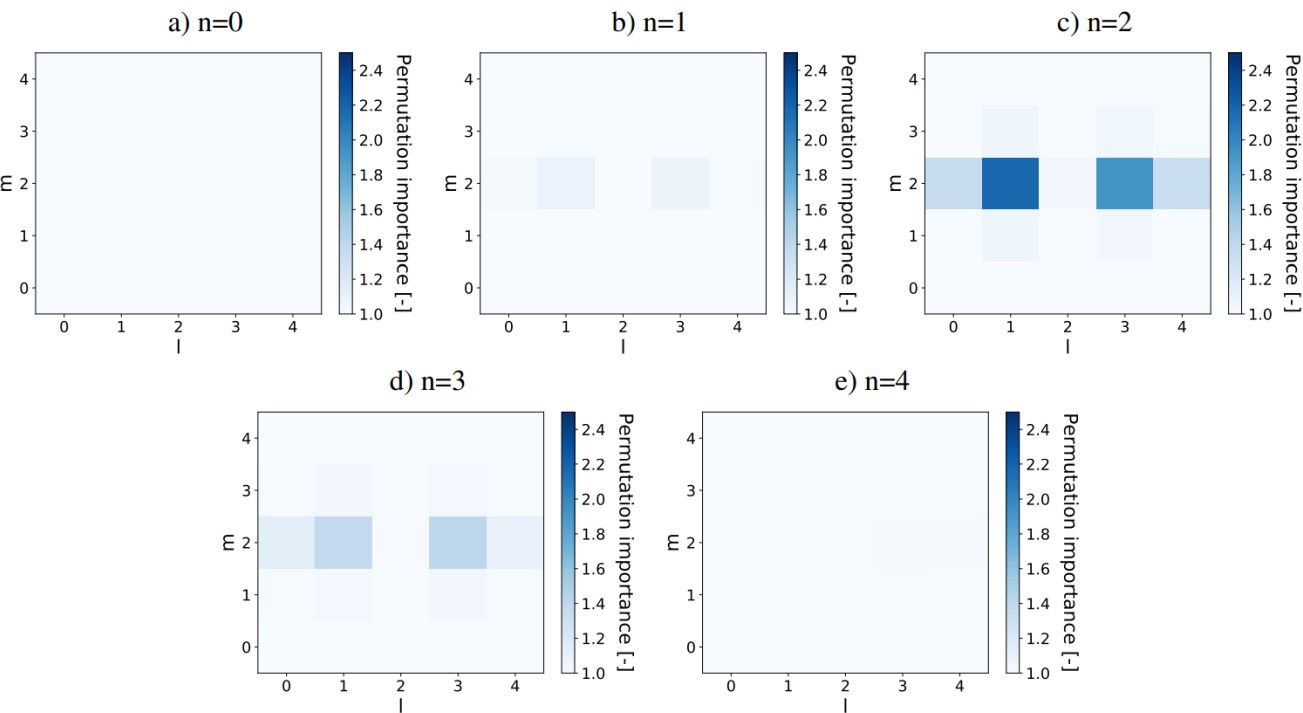

**Figure 14.** Permutation feature importance of all $u$-velocities in the local input stencil (with indices $(l, m, n)$) associated with predicting $\tau_{wu}^{out}$ in the log-layer ($0.09375\frac{z}{\delta}$ ($55.3125z^+$)) using an ANN (with $n_{hidden} = 64$) trained and tested only on $f_{hor} = 8$, averaged over 3 flow fields reserved for a priori testing and 10 random shufflings. The five panels a-e each show one of the five horizontal planes (indicated by their vertical index $n$) present in the input stencils. $\tau_{wu}^{out}$ is located in the center of the shown horizontal plane, halfway between $n = 2$ and $n = 3$.

## 4.2 A posteriori (online) test

Our ANN LES SGS model produced numerically unstable a posteriori results without resorting to artificially introducing
additional variance (for instance via eddy-viscosity models) or imposing strong ad-hoc numerical constraints, which is in agreement with the results of Beck et al. (2019), Maulik et al. (2019), and Zhou et al. (2019).

Several other studies (Guan et al., 2021; Park and Choi, 2021; Wang et al., 2018; Xie et al., 2019; Yang et al., 2019), in contrast, did report stable a posteriori results without requiring ad-hoc adjustments, although in some cases only after using single-point rather than multi-point inputs (Park and Choi, 2021), or ensuring that sufficient training samples are presented
(Guan et al., 2021).

We emphasize though that all the aforementioned studies (with the notable exception of Park and Choi (2021)), do not consider wall-bounded flows. In addition, they do not compensate for the instantaneous spatial discretization errors associated with a staggered finite-volume grid.

Crucially, for our set-up these spatial discretization errors were substantial, removing a large part of the variance present at high wave modes in the DNS (Fig. 2). Since we designed our ANN SGS model to fully compensate for these instantaneous discretization errors, our SGS model tended to re-introduce a large amount of variance at the highest resolved wave modes. In the a priori test, we consistently found that the Smagorinsky SGS model, as opposed to our ANN SGS model, strongly underestimated the small-scale variability of $\tau_{ij,DNS}$ and $\epsilon_{SGS,DNS}$ (Sect. 4.1.2).

The introduction of additional variance at the highest wave modes by our ANN SGS model is on its own not necessarily a problem if the energy transfer from the resolved to the unresolved scales is sufficient. Our ANN SGS model, consequently, needs to provide sufficient additional dissipation, compared to the SGS models from the aforementioned studies and traditional SGS models like Smagorinsky. Promisingly, a priori we found that the ANN matched well the net dissipation inferred from the DNS (Fig. 11), and indeed provided more net dissipation than the traditional Smagorinsky SGS model. Despite that, we observed a posteriori a gradual pile-up of spectral energy at the smallest wave modes (shown as an example for the $u$-component in Fig. 15), indicating that an overall lack of dissipation nonetheless remains.

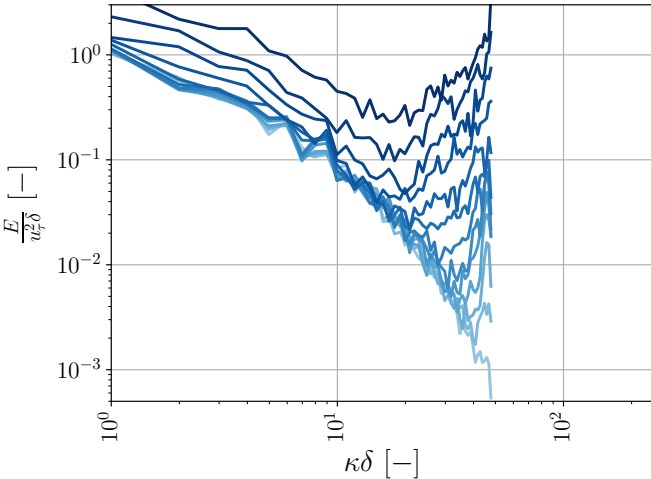

**Figure 15.** Time evolution of stream-wise spectra averaged in the span-wise direction, where the colour brightness indicates the different time steps. Here, the light blue colours refer to the first time steps, and the dark blue colours to the final time steps. The time steps range from t=0s to t=36s, with intervals of 3s.

We hypothesize that two issues prevented the ANN SGS model from producing the required dissipation a posteriori: 1) error accumulation, and 2) aliasing errors.

In the first place, similar to Beck et al. (2019), we hypothesize that high-frequency errors in the ANN predictions accumulated over time due to strong positive feed-backs between our ANNs and the LES simulation. We stress that ANN SGS models like ours can never be perfect, and consequently will always introduce errors in an a posteriori simulation that affect, together with the full LES dynamics, its own inputs in the next time step. It strongly depends on the characteristics of the SGS model, whether this can result in positive feedback loops that cause divergence from the physical solution and subsequent numeric instability.

In this regard, eddy-viscosity models like Smagorinsky have an important stabilizing property in steady-state channel flow: as soon as the energy content starts deviating from the physical solution, the subgrid dissipation is automatically adjusted (via a change in the gradients serving as input) to compensate for it.

Such a stabilizing property, however, was clearly lacking in our ANN SGS model. This is not surprising: we designed our ANN SGS model to compensate for many spatial discretization effects, which typically dampen the error accumulation at high-frequencies. It is well-known that, due to the chaotic nature of turbulence, small errors introduced by the predicted transports have a tendency to grow over time (e.g. Liu et al., 1994). On top of that, it has been shown before by Nadiga and Livescu (2007) that 'perfect' SGS models (that exactly compensate for the unresolved physics, modelling errors, and instantaneous discretization errrors), are inherently unstable in implicit-filtering LES due to the presence of multiple different attractors. These issues were likely exacerbated by the growing need for the ANN to extrapolate beyond its training state once the simulation started deviating from the physical solution. This extrapolation likely increased the ANN prediction errors, which would in turn accelerate the divergence from the physical solution.

In the second place, we hypothesize that, during the a posteriori test, aliasing errors became prominent due to the introduced variance at high wave modes. Such aliasing errors are known to introduce instability when not dampened by discretization errors and/or dealiasing techniques (e.g. Kravchenko and Moin, 1997; Chow and Moin, 2003). The quadratic velocity products in the non-linear advection term, can in principle introduce wave modes that are not supported by the finite LES grid. The additional variance could have prevented them from being dampened by the instantaneous spatial discretization errors, causing them to appear as spurious resolved wave modes in the finite LES solution. This would increase the amount of dissipation required in the LES simulation. These aliasing errors were not accounted for during the ANN training, as it only relied on *instantaneous* coarse-grained flow fields that did not contain additional variance.

## 5 Conclusions and recommendations

In this study, we evaluated and developed a data-driven large-eddy simulation (LES) subgrid-scale (SGS) model based on artificial neural networks (ANNs) that aims to represent both the unresolved physics and instantaneous spatial discretization errors. We focused specifically on the widely-used LES approach where a staggered finite-volume grid acts as an implicit filter, where the discretization errors can strongly interact with the resolved physics.

We designed our ANN SGS model such that, similar to conventional eddy-viscosity SGS models like Smagorinsky, it can be applied locally in the grid domain: the employed ANNs used as input only local $5 \times 5 \times 5$ stencils of the resolved wind velocity components $(\overline{u}, \overline{v}, \overline{w})$. Interestingly, an additional analysis we performed with so-called permutation feature importances, suggested that our ANNs mostly focused on a small part of the stencils oriented along the mean flow direction. Hence, the input stencils we used could perhaps be further optimized by selecting smaller stencils that extend along the mean flow direction.

Using as a test case turbulent channel flow (with $Re_\tau = 590$), we trained the ANNs with individual 3D flow fields obtained from direct numerical simulation (DNS). By applying an explicit finite-volume filter (i.e. a discrete 3D top-hat filter) on the

high-resolution DNS fields and mimicking the instantaneous spatial discretization errors made in actual LES, we generated millions of ANN input-output pairs that allowed us to train the ANNs in a supervised manner.

Subsequently, we performed both an *a priori, offline* and *a posteriori, online* test. As an a priori test, we directly compared the ANN predictions to the DNS derived values for flow fields unseen during training. Focusing first on the relatively simple case where a single coarse horizontal coarse resolution is used during both training and testing, we found, in general, excellent agreement for all heights in the channel: the spatial patterns in the DNS values were well captured, and the correlation coefficients between the ANN predictions and DNS values were high (mostly between $0.6$ and $1.0$). For a single coarse resolution, the ANNs were thus well able to represent the unresolved physics and instantaneous spatial discretization errors in the entire flow, based only on the resolved flow fields. We do note that we did find a few shortcomings that can possibly be improved upon: the extreme SGS fluxes were slightly underestimated, and the predicted horizontally averaged vertical profile of $\tau_{wu}$ deviated in particular close to the walls.

In addition, we tested the generalization performance of our ANN SGS model with respect to the selected coarse horizontal resolution. We found that the ANN could be successfully trained on multiple resolutions simultaneously, and was in most cases able to generalize to other resolutions unseen during training. The generalization performance was particularly good when the unseen resolution was within the range of the resolutions seen during training, suggesting that a limited set of training resolutions may be sufficient to achieve a good generalization performance with respect to the selected resolution. The generalization performance of our ANN SGS model towards other flow types and/or higher Reynolds number though, is currently still an open issue. This can possibly be overcome by applying known scalings and properties to the inputs and outputs (e.g. Ling et al., 2016b; Yang et al., 2019), extending the range of cases covered in the training data-set, and/or retraining a previously optimized ANN on limited data from a new flow through transfer learning (Guan et al., 2021).

To test our ANN SGS model a posteriori, we incorporated a trained ANN SGS model directly into an actual LES of the selected turbulent channel flow test case. Contrary to the a priori test, the ANN SGS model did not produce satisfactory results. Since our ANN SGS model, in contrast to traditional SGS models like Smagorinsky, compensated for many spatial discretization effects by introducing additional variance, the need for additional dissipation increased. The ANN SGS model appeared not to provide this dissipation sufficiently, causing an artificial build-up of TKE at high wave-modes that eventually destabilized the solution. We hypothesized that our ANN SGS model did not produce sufficient dissipation because of 1) error accumulation, and 2) aliasing errors.

We thus conclude that our ANN SGS model cannot, in its current form, achieve computationally stable results without resorting to previously suggested ad-hoc adjustments (e.g. neglecting all backscatter or combining with the Smagorinsky SGS model). These ad-hoc adjustments, however, (re-)introduce strong assumptions, and obscure the link between the a priori and a posteriori SGS model. We therefore would like to mention below a couple of possible alternative approaches, which may help to circumvent the need for ad-hoc adjustments and could therefore be worth exploring further in future studies.

First of all, one way forward could be to adjust the training procedure such that it reflects better the a posteriori simulation. A potential elegant way to achieve this may be an *online* learning procedure similar to the ones proposed by Rasp (2020) and Guan et al. (2021), where the ANN SGS model would be trained within the actual, online LES simulation to reproduce the

correct statistics and/or the correction terms inferred from a dynamically coupled DNS. Alternatively, errors expected to be introduced by an ANN SGS model in an a posteriori LES simulation, could be added offline to the filtered flow fields $\overline{u}, \overline{v}, \overline{w}$ during training. This may help to reduce the sensitivity of the ANN to its own errors, alleviating the need for extrapolation once the a posteriori LES simulation starts diverging.

A second way forward may be to further improve the design of our data-driven SGS model, in particular including more physical constraints and insights. It could for instance be interesting to include the SGS transfer terms in the loss function used during offline training, as it may allow the ANNs to improve their representation of the net SGS transfer. This could make the ANN SGS model less prominent to a posteriori instability.

Besides that, it is likely worthwhile to further optimize the chosen inputs, the selected machine-learning algorithm, and the training when attempting to stabilize the a posteriori simulation. Park and Choi (2021) found for instance that using single-point inputs, rather than multi-point inputs, alleviated the observed a posteriori instability. Guan et al. (2021) observed that so-called *convolutional* neural networks achieved higher a priori accuracy than the multi-layer perceptron architecture selected in this study, and, interestingly, that the a posteriori stability depended on the number of training samples. Possibly, a corresponding further increase in the a priori accuracy helps to reduce the error accumulation a posteriori, making the a posteriori simulation less prone to instability.

All in all, our developed ANN LES SGS model has, based on its excellent a priori performance, potential to improve the representation of the unresolved physics and discretization errors in turbulent flows. However, the developed ANN LES SGS model is in its current form still prone to numeric instability in a posteriori simulations. Hence, several open challenges remain before the potential of ANN LES SGS models like ours can be successfully leveraged in practical applications.

*Code availability.* All the code used to generate the data and figures in this paper, including the employed CFD code MicroHH (van Heerwaarden et al., 2017a, b), are hosted at a GitHub repository located at https://github.com/robinstoffer/microhh2/tree/NN. In addition, the used repository has been archived at http://doi.org/10.5281/zenodo.4767902 (Stoffer, 2021). To facilitate a posteriori LES simulations with the ANN SGS model developed in this paper, the parameters of the ANN selected in this paper are included as txt-files.

*Author contributions.* RS generated the training data, wrote the ANN SGS model, and tested its performance in an a priori and a posteriori
test. CvH supported RS in developing and writing the ANN SGS model. RS and CvH developed the ideas that led to the study. CvH, CvL, DP, VC, and MV all provided extensive feedback on the training data generation, ANN design, and hard-ware specific tuning. MJ helped in interpreting the a posteriori results and coming up with suggestions for future research. RS wrote the manuscript. CvH, OH, CvL, MJ, and MV provided valuable feedback on the manuscript.

*Competing interests.* The authors declare that they have no conflict of interest.

*Acknowledgements.* The authors gratefully acknowledge the SURF Open Innovation Lab and WIMEK graduate school (WUR) for funding this study.

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
