# Peer review of "Development of a large-eddy simulation subgrid model based on artificial neural networks: a case study of turbulent channel flow"

_Geoscientific Model Development, 2020_

## Referee Comment (RC1) · Anonymous Referee #1 · 17 Dec 2020

**1   General comments**

The manuscript "Development of a large-eddy simulation subgrid model based on artificial neural networks: a case study of turbulent channel flow" (gmd-2020-289) by Stoffer et al. describes the training and testing of an artificial neural network (ANN) intended to act as a a large-eddy simulation (LES) subgrid-scale (SGS) turbulence closure model. The ANN was trained on filtered direct numerical simulation (DNS) fields, then compared to the popular Smagorinsky-Lilly eddy-viscosity model in *a priori* and *a posteriori* applications to mock LES fields (previously unseen filtered DNS fields). The ANN SGS model performs well in the *a priori* case where it is applied offline to a

filtered DNS field, with its predicted turbulent stresses matching those filtered from the DNS much closer than the Smagorinsky-Lilly model both visually in the spatial patterns of the predicted stresses and in their probability distribution and spectrum. The ANN was then implemented in their DNS/LES code (MicroHH) as an LES turbulence closure model for the *a posteriori* test, but was unable to dissipate enough fine-scale energy to maintain a stable simulation.

The manuscript is well-referenced and thoroughly rigorous in its development, and their code/parameters are made publicly available for reproduction which is appreciated. However, the experiment and results presented are not a significant contribution to the literature for a few reasons. Most notably, the ANN was not successful as an LES SGS turbulence closure while being 15-fold more expensive to run than the Smagorinsky-Lilly model, not including the cost of training the ANN or producing the training DNS fields. Further to this point (and as they note), their result that the ANN model generates too much backscatter and not enough dissipation has been both seen and addressed with different methods in the literature. The ability of the ANN to predict accurate SGS stresses based on filtered velocity fields in an *a priori* setting and the analysis of which input fields the ANN deems important to achieve these accurate results is an interesting avenue which could be a valuable contribution to the turbulence-modeling literature, but it is not pursued deeply enough currently to warrant publication.

Recommendation: Major revisions

**2 Specific comments**

**2.1 Focus is too broad and unbalanced**

The presentation is divided relatively evenly between the methodology for filtering the DNS fields for the specific finite-volume filter including numerical errors, the training of

the ANN, the ANN-produced *a priori* stress fields, the permutation feature importance of the ANN in the *a priori* experiment, and the *a posteriori* results and potential explanations of the observed instability. The result is a very broad outline of the developed ANN SGS turbulence closure with a long description of the methodology and relatively sparse analysis of the results, which would be more appropriate if the model was completely novel or more successful as a functional LES turbulence closure, or both. I recommend that the authors pick one aspect to focus on and thoroughly analyze for this submission and potentially revisit the others separately. The description of the finite volume filtering is particularly verbose with ten full lines dedicated to equations and could be greatly reduced unless the filtering process is decided to be the focus of the manuscript.

**2.2 Only one test case**

The ANN is trained on fields from a single neutral DNS case at steady state then evaluated on steady-state fields from the same DNS case. The performance of the ANN in the *a priori* test would be much more striking if it were demonstrated that it was able to accurately model SGS stresses for a case that it was not trained on, or even if it was still evaluated for a case it was trained with but shown that the ANN maintains its performance when trained on multiple cases. Put another way, it is not clear here if the ANN is learning about turbulence in general or about a single steady-state field specifically, which makes the results difficult to properly digest. The potential insights into turbulence modeling from the analysis of permutation feature importance could be quite interesting if it is shown that there is some generality to the ANN.

**2.3  Unstable when used as a live model**

The result that the ANN SGS model leads to an unstable LES solution is not worthy of publication. I suggest that the authors either re-focus the manuscript on the implications of what can be learned from the *a priori* results or implement some of the possible solutions that they mention (e.g. limiting backscatter, adding dissipation via an eddy-viscosity component) to achieve numerical stability and discuss the implications of the amount of tuning necessary to, for example, their unique filtering process including numerical errors or the general formulation of mixed models (which are often very ad hoc and could use insights from new sources).

**2.4  Minor comments**

- A formulaic description of your implementation of the Smagorinsky-Lilly model and its associated wall model would be helpful

- Table 3 would be more digestible as a figure

- Dissipation ($-\tau_{ij}S_{ij}$) would be nice to see in the analysis of the *a priori* results, particularly given the low energy in the Smagorinsky-Lilly results for just $\tau_{ij}$ and the *a posteriori* outcome for the ANN

- Line 206: "Simply boils down to..." is overly casual

**3  Technical corrections**

- Fig. 2: The titles on the individual fields are much too small to read at 100% resolution

- Line 366: "For the u-input stencil the $u$-velocity input stencil"

---

## Referee Comment (RC2) · Noah Brenowitz (Referee) · 17 Feb 2021

**General Comments**

This manuscript applies ML to turbulence closures in idealized channel simulations with Large Eddy Simulations (LES). The authors compute the turbulent stress tensor from a direct numerical simulation with 8x the resolution of the LES. They coarse-grain the frictional stresses and momentum fluxes through interfaces that follow the staggered grid of the LES, and compute the turbulent portion as the residual from the LES-resolved fluxes computed via interpolation. NNs trained to predict this stress tensor dramatically outperform a traditional Smagorinsky LES closure, and have otherwise reasonable skill

"a priori". However, they do not have stable results when the NNs are coupled to the LES (i.e. "a posteriori").

The paper is nicely presented and thorough. I enjoyed reading it. It is disappointing that they didn't achieve good "a posteriori" skill, but this is hard to do, and I appreciated their discussion of the issues. I recommended the paper be accepted, but I would do have some comments:

1. The introduction has a comprehensive literature review of ML for turbulence closures in the CFD/engineering community, but could also mention similar work in the oceanography/atmospheric science on e.g. geostrophic turbulence (Bolton and Zanna, 2019), boundary layer parameterization (McGibbon and Bretherton, 2019), and "full-physics" parameterization (Brenowitz and Bretherton, 2019; Rasp et. al. 2018; Yuval and O'Gorman 2020). These papers all formulate similar coarse-graining problems, and I think readers would enjoy seeing the connections.

2. The question of "a posteriori" and "a priori" errors is a key unsolved problem in the literature. Overall this study diligently accounted for the spatial discretization of the LES, but did not handle the time discretization. For example, would decreasing the time step or using a different time stepper stabilize the "a posteriori" simulations, or is the ML problem they have posed intrinsically unstable?

**Specific Comments**

L55: "a posteriori tests"

This jargon has diverged somewhat in the turbulence and weather/climate modeling communities. Climate modelers frequently use the terms "online" and "offline" to denote a posteriori and a priori validations, respectively. It would be good to point this out for the geoscience reader.

L95: "Friction Reynold's number"

I assume this refers to the Reynold's number arising from molecular viscosity, but it

would be clearer if it were defined in terms of the geometry, viscocity etc in Eq 1.

Also, it should be pointed out that atmospheric flows have orders of magnitude higher Reynold's numbers than Re=590.

L185 (Fig 2). Please define "friction velocity".

L325. How important was this preferential sampling in terms of quantitative performance?

Figure 8. The Smagorinsky scheme does dramatically worse. I am concerned this is a straw-man comparison. Is Smagorinsky a state-of-the-art baseline for this problem?

L415. These issues were likely exacerbated by the growing need for the ANN to extrapolate beyond its training state once the simulation started deviating from the physical solution.

Many ML-assisted weather models show identical problems. These spiraling errors are can be revealed by normal mode analysis of the underlying linearized ML-fluid system (see Brenowitz, et. al. 2020). It would be fascinating to see this technique applied to NNs here in some future paper.

**Technical Corrections**

L83: "instantaneuous" is misspelled

Figure 3. This graphic is busy, but I understood what they meant from the text.

Figure 9. This graphic has small text and shows some over-plotting in Fig 9a. I suggest further reducing the transparency of the markers or using a hexbin plot.

Line 340. This section header needs a preposition e.g. "Comparison *to* Smagorinsky"

**References**

Bolton, T. & Zanna, L. Applications of Deep Learning to Ocean Data Inference and Sub-Grid Parameterisation. J. Adv. Model. Earth Syst. (2019) doi:10.1029/2018MS001472

Rasp, S., Pritchard, M. S. & Gentine, P. Deep learning to represent subgrid processes in climate models. Proc. Natl. Acad. Sci. U. S. A. 115, 9684–9689 (2018)

Brenowitz, N. D. & Bretherton, C. S. Spatially Extended Tests of a Neural Network Parametrization Trained by Coarse‐graining. J. Adv. Model. Earth Syst. (2019) doi:10.1029/2019MS001711

Brenowitz, N. D., Beucler, T., Pritchard, M. & Bretherton, C. S. Interpreting and Stabilizing Machine-learning Parametrizations of Convection. Journal of Atmospheric Sciences Early online release, (2020)

McGibbon, J. & Bretherton, C. S. Single‐Column Emulation of Reanalysis of the Northeast Pacific Marine Boundary Layer. Geophys. Res. Lett. (2019) doi:10.1029/2019GL083646

Yuval, J. & O'Gorman, P. A. Stable machine-learning parameterization of subgrid processes for climate modeling at a range of resolutions. Nat. Commun. 11, 3295 (2020)

---

## Author Response (AR2)

**Final author response on: "Development of a large-eddy simulation subgrid model based on artificial neural networks: a case study of turbulent channel flow (gmd-2020-289)"**

Robin Stoffer et al.

April 14, 2021

*Edit 14-4-2021: Where applicable, we will indicate below each reply in our original rebuttal submitted one month ago, in an italic font, the most relevant changes we eventually made in the revision. We note that this does not include all the changes we have made, given the extensive nature of the revision.*

In this author comment, we will respond to the comments of the two referees. First of all, we would like to thank both of them, together with the editor, for the time and effort they have invested in the peer review of our manuscript. The comments of both referees are in our view very valuable, and help us a lot in improving the quality of our manuscript.

Below, we will give our response to the individual comments of the referees. Each time, we will first show the corresponding quote from the review in blue, and subsequently our response. However, before we respond to the individual comments, we first would like to re-clarify the scope and intentions of our paper, as we realized from the reviews that this requires improvement, and we would like to motivate why we see a need in documenting the observed a posteriori instability.

**1 Intended scope and novelty of paper**

In particular, the concerns raised by referee #1 made clear to us that we need to motivate the intentions and novelty of our paper better.

The intended message of our paper was two-fold. First, we aimed to thoroughly document our novel training data generation procedure that incorporates both the subgrid physics and discretization errors. Second, using one of the simplest wall-bounded flow cases (which is already a more complicated case than most other ANN SGS modelling studies have considered), we wanted to make a first assessment whether an ANN SGS model trained on our novel training dataset, can potentially be implemented in an actual LES without having to resort to previously used ad-hoc adjustments (e.g. by combining it with Smagorinsky). We therefore opted to limit the scope of the a priori test, and instead provide a discussion of the currently observed a posteriori instability, together with suggestions for future research that may circumvent the need for the previously used ad-hoc adjustments. We emphasize that the previously used ad-hoc adjustments to tackle the instability, are not ideally preferred solutions with important issues (e.g. ad-hoc tuning, re-introduction assumptions of Smagorinsky SGS model, neglecting back-scatter), and we hope that documenting the observed instability will help the field to assess alternative, possibly better approaches.

We nonetheless understand the concern of referee #1 that the a priori test, in its current form, does not sufficiently test the generalizability of our ANN SGS model, especially as we only demonstrated the ANN performance for one coarse resolution.

We will revise the paper in order to ensure that our intentions come better across, in particular in the introduction and conclusion sections. In addition, we will extend the a priori analysis by training and testing our ANN SGS model on multiple horizontal coarse resolutions simultaneously, including tests on horizontal coarse resolutions unseen during training.

*Edit 14-4-2021: We revised the introduction and conclusion sections as stated. In particular, at the end of the introduction we now explicitly emphasize the intentions mentioned above (lines 97-109). In addition, we included the results of the described additional analysis in a new Sect. 4.1.3. We explain the methodology behind the additional analysis in Sect. 3.5.1.*

**2 Response to comments of Referee #1**

**2.1 general comments**

*"The manuscript is well-referenced and thoroughly rigorous in its development, and their code/parameters are made publicly available for reproduction which is appreciated."*

We thank for the referee for these kind words. We hope that the thorough description of our approach and the remaining challenges, together with the used code, will contribute to the development of accurate, stable, and generalizable ANN SGS models.

*"However, the experiment and results presented are not a significant contribution to the literature for a few reasons. Most notably, the ANN was not successful as an LES SGS turbulence closure while being 15-fold more expensive to run than the Smagorinsky-Lilly model, not including the cost of training the ANN*

*or producing the training DNS fields."*

We do believe that our experiment and results are a significant contribution to the literature. Most importantly, we thoroughly document an ANN SGS model that has potential to accurately compensate for both subgrid physics and discretization errors for a staggered finite-volume grid. As mentioned in Sect.1, we hope that documenting the observed a posteriori instability will help the field to overcome it without resorting to ad-hoc adjustments/constraints, allowing to fully leverage the observed a priori potential (see also Sect.2.2).

Regarding the computational cost, an actual LES with our ANN SGS model (with $n_{hidden} = 64$) is indeed about a factor 15 slower than an equivalent LES with the Smagorinsky SGS model (as we also mention in our manuscript). We do note that we focused mostly on testing the potential of our approach, rather than optimizing (yet) its computational cost. Hence, the computational cost of our ANN SGS model can still be further improved when required.

In addition, we note that the Smagorinsky SGS model is computationally already very efficient, and thus the ANN cannot be expected to outperform the Smagorinsky SGS model in this respect. In line with more advanced SGS models like scale-dependent dynamical SGS models, the primary objective of ANN SGS models like ours is instead to (eventually) improve the accuracy of LES for a given resolution (and in doing so may allow a computationally cheaper lower resolution to achieve a certain desired level of accuracy). We will better emphasize this point in the methodology sections.

*Edit 14-4-2021: We eventually clarified the point mentioned above in the introduction section rather than the methodology sections. Specifically, we clarified it at lines 98-101.*

*"Further to this point (and as they note), their result that the ANN model generates too much backscatter and not enough dissipation has been both seen and addressed with different methods in the literature."*

We are indeed aware of some papers that documented a posteriori instability in the context of ANN SGS modelling, and subsequently came up with ad-hoc adjustments (e.g. artificially introducing dissipation by combining with Smagorinsky) and/or strong numerical constraints (e.g. eliminating back-scatter) that alleviated the instability. We mention and discuss these papers in the a posteriori section, together with other papers that, interestingly, did not document any instability when using ANN SGS models a posteriori. If the referee referred to other literature than the papers we cited, we would appreciate it if he/she provides the references.

Despite that some papers indeed mentioned the a posteriori instability, as mentioned before we still think it is important to document the a posteriori instability we encountered. We will elaborate more on the reasons why in Sect.2.2.

*"The ability of the ANN to predict accurate SGS stresses based on filtered velocity fields in an a priori setting and the analysis of which input fields the ANN deems important to achieve these accurate results is an interesting avenue which could be a valuable contribution to the turbulence-modeling literature, but it is not pursued deeply enough currently to warrant publication."*

We agree that exploring in more depth the permutation feature importances, in combination with a more extensive a priori test, would be very interesting, as it may provide new insights in the dominant locally resolved flow patterns that drive subgrid physics and numerical errors. We decided to extent the current a priori test with ANNs trained on multiple horizontal coarse resolutions simultaneously (see Sect.1 and 2.2). This allows us to explore whether the permutation feature importances are influenced by the coarse horizontal resolutions seen during training. If we observe that the permutation feature importances are indeed substantially influenced, we will add our findings to the permutation feature importance section in the manuscript.

*Edit 14-4-2021: We reflected the new results in the section discussing the permutation feature importance results (which is in the revised manuscript Sect. 4.1.4.). In particular, we now mention two additional interesting findings we did not mention in the previous version (lines 462-465).*

**2.2 Specific comments**

In this section, we will again respond to each comment individually. However to keep things concise, where appropriate, we will refer to our responses to the general comments and our general statement about the intended scope of the paper.

*"2.1 Focus is too broad and unbalanced*

*The presentation is divided relatively evenly between the methodology for filtering the DNS fields for the specific finite-volume filter including numerical errors, the training of the ANN, the ANN-produced a priori stress fields, the permutation feature importance of the ANN in the a priori experiment, and the a posteriori results and potential explanations of the observed instability. The result is a very broad outline of the developed ANN SGS turbulence closure with a long description of the methodology and relatively sparse analysis of the results, which would be more appropriate if the model was completely novel or more successful as a functional LES turbulence closure, or both. I recommend that the authors pick one aspect to focus on and thoroughly analyze for this submission and potentially revisit the others separately. The description of the finite volume filtering is particularly verbose with ten full lines dedicated to equations and could be greatly*

*reduced unless the filtering process is decided to be the focus of the manuscript."*

In retrospect, we understand the view expressed here by the referee. It made us realise that we did not convey clearly enough the intentions and novelty of the manuscript (see Sect.1), and, in addition, the methodology sections were too detailed on some occasions. We will therefore attempt to shorten the methodology sections by focusing it more on the novel aspects (in particular the finite-volume filter) and/or moving parts of the methodology sections to an appendix.

Besides that, we plan to extend the analysis of the ANN performance in the a priori test (see next response and Sect.2.3) such that, overall, the balance between the methodology and results sections shifts more towards results.

*Edit 14-4-2021: We extensively revised the manuscript to alleviate this point raised by the referee. Most importantly, we shortened several methodology sections and restructured the manuscript to improve readability. We removed two long formulas in Sect. 2 (i.e. the long defintions of $\tau_{ij}$), strongly reduced Sect. 3.2 by referring to Sect. 2 (which includes the removal of two detailed methodology figures present in the original manuscript), and removed several explanations and details in Sect. 3.4 (removing, amongst others, an explanation about backpropagation). We restructured the paper to make a more clear distinction between the theoretical framework we introduce (Sect. 2), our methodology sections (Sect. 3), and the results (Sect. 4). Here, we moved the long introductions of the original a priori and a posteriori sections to a new Sect. 3.4. In addition, at the beginning of the new Methodology and Results sections we give an overview of the discussed content (lines 176-181, 352-356).*

*"2.2 Only one test case*

*The ANN is trained on fields from a single neutral DNS case at steady state then evaluated on steady-state fields from the same DNS case. The performance of the ANN in the a priori test would be much more striking if it were demonstrated that it was able to accurately model SGS stresses for a case that it was not trained on, or even if it was still evaluated for a case it was trained with but shown that the ANN maintains its performance when trained on multiple cases. Put another way, it is not clear here if the ANN is learning about turbulence in general or about a single steady-state field specifically, which makes the results difficult to properly digest. The potential insights into turbulence modeling from the analysis of permutation feature importance could be quite interesting if it is shown that there is some generality to the ANN."*

This is a valid point raised by the referee. Rather than performing such an extensive a priori test, we opted in this manuscript to test whether the developed ANN SGS model could be used in an actual LES without ad-hoc adjustments (see Sect.1). Since we selected for the a posteriori test the same LES case as represented during the training and a priori test, we could assess this question

already with the a priori test in its current form. In addition, the current a priori test does show the capability of the ANN to accurately predict the subgrid physcis and numerical errors for previously unseen realizations of the selected channel flow case.

As mentioned in Sect.1, we do understand the point of the referee that we did not sufficiently demonstrate the generalizability of our ANN SGS model, mainly because we selected only one coarse resolution. We will therefore add an additional sub-section to the a priori test, where we will assess to what extent our ANN SGS model can be simultaneously trained on two/three different horizontal coarse resolutions, and generalize towards horizontal coarse resolutions unseen during training. To this end, using three different coarse horizontal resolutions (with horizontal coarsening factors of 4, 8, and 12 resp.), we will train and test our ANN SGS model in three different ways:

1. Train only on a coarsening factor of 8 (which is equivalent to how we currently trained our ANN), test on all three coarsening factors.

2. Train on coarsening factors of 4 and 12, test on coarsening factor 8.

3. Train on all three coarsening factors simultaneously, test on all coarsening factors.

Given the current scope of the paper (Sect.1), we do still intend to exclude the generalization towards higher Reynolds number and other flow types (e.g. pipe flow, unstable/stable wall-bounded flows). In our view, doing such an assessment properly would warrant a separate publication. As we focus in our manuscript on incorporating numerical errors in the training data, we think the additional analysis we propose above is most relevant for the current scope of the paper. In the conclusions section, we will additionally mention that the generalizability of our ANN SGS model towards other Reynolds numbers and flow types is still an open issue.

*Edit 14-4-2021: We included the suggested additional analysis. We explain the methodology behind the analysis in Sect. 3.5.1, and discuss the results in Sect. 4.1.3. We discuss the implications in the abstract and conclusions sections. We also emphasize the generalization assessed in the original analysis (lines 285-287). Finally, as stated we included a note in the conclusion section that the generalization towards other Re and flow types remains an open issue, together with directions for future research (lines 557-560).*

*"2.3 Unstable when used as a live model*

*The result that the ANN SGS model leads to an unstable LES solution is not worthy of publication. I suggest that the authors either re-focus the manuscript on the implications of what can be learned from the a priori results or implement some of the possible solutions that they mention (e.g. limiting backscatter, adding dissipation via an eddy-viscosity component) to achieve numerical stability and discuss the implications of the amount of tuning necessary to, for*

*example, their unique filtering process including numerical errors or the general formulation of mixed models (which are often very ad-hoc and could use insights from new sources)."*

In contrast to the reviewer, we do believe that the observed a posteriori instability is interesting for the readers of our paper. In our view, there are three important reasons why it is valuable to document and emphasize the currently observed a posteriori instability:

1. The ad-hoc adjustments and/or strong numerical constraints used by several previous papers (Beck et al., 2019; Maulik et al., 2019; Zhou et al., 2019) to resolve the issue, are not ideally preferred, as they require e.g. ad-hoc tuning, re-introduction of the assumptions associated with the Smagorinsky SGS model, and/or neglecting back-scatter. This is the main reason why we think it is more worthwhile to document and discuss the observed a posteriori instability, rather than implementing these previously used adjustments/constraints. In the conclusions section, to this end we also highlight several other alternative approaches that may help to circumvent the need for the proposed ad-hoc adjusments/constraints completely.

2. There does not seem to be a consensus in fluid mechanics literature whether and when the a posteriori instability occurs. Several papers (Wang et al., 2018; Yang et al., 2019; Xie et al., 2019) did not report any instability when testing their ANN SGS models a posteriori.

3. Potentially, many GMD readers may experience similar issues in different contexts, highlighting interesting cross-disciplinary connections. As the second referee mentions, the problem that an ANN parameterization performs well in an a priori (or "offline") setting, but badly in an a posteriori (or "online") setting, is also seen in other geoscientific disciplines like climate modelling (Brenowitz and Bretherton, 2019; Rasp, 2020, e.g.).

We will better emphasize the issues associated with the previously used methods to tackle the instability, and the alternative solutions we propose (in particular in the conclusions section). In addition, we will also mention the findings of two very recent publications (Park and Choi, 2021; Guan et al., 2021), which, similar to our paper, also highlight the a posteriori instability encountered with ANN SGS modelling. Guan et al. (2021) also cites our paper in this context.

*Edit 14-4-2021: We emphasize the issues with the previously used ad-hoc methods now both in the introduction (lines 59-61) and the conclusion (lines 570-571). In addition, we rewrote the conclusions sections, where we now emphasize more the alternative solutions we propose (lines 571-590). We also*

*emphasize this in the revised abstract. We mention some findings of the two recent studies in the introduction (lines 73-76), the a posteriori test in Sect. 4.2 (lines 482-485), and the conclusions (amongst others lines 585-588).*

**2.3   Minor comments**

*"A formulaic description of your implementation of the Smagorinsky-Lilly model and its associated wall model would be helpful"*

Agree. We will include it.

   *Edit 14-4-2021: We included it at lines 298-306.*

*"Table 3 would be more digestible as a figure"*

Agree. We will plot the correlation coefficient as vertical profiles instead.

   *Edit 14-4-2021: We show the corresponding new plot in Fig. 12.*

*"Dissipation ($\tau_{ij}S_{ij}$) would be nice to see in the analysis of the a priori results, particularly given the low energy in the Smagorinsky-Lilly results for just $\tau_{ij}$ and the a posteriori outcome for the ANN"*

Agree, this is indeed a very good suggestion, especially given the apparent lack of dissipation in the a posteriori test. We will calculate and include the dissipation in the a priori test. If appropriate, we will also discuss its implications for the observed a posteriori instability.

   *Edit 14-4-2021: We now explain the dissipation in Sect. 3.5.1., and included the results in Sect. 4.1. We discuss it in the context of the a posteriori outcome at lines 385-388 and 497-500.*

*"Line 206: "Simply boils down to..." is overly casual"*

Agree, we will rephrase it.

   *Edit 14-4-2021: In shortening the manuscript, we removed the sentence corresponding to this comment.*

**2.4   Technical corrections**

*"Fig. 2: The titles on the individual fields are much too small to read at 100% resolution"*

Agree. We will enlarge the titles of the individual fields, such that they are better readable.

*Edit 14-4-2021: In shortening the manuscript, we removed the figure corresponding to this comment.*

**3 Response to comments of Referee Noah Brenowitz**

**3.1 General comments**

*"The paper is nicely presented and thorough. I enjoyed reading it. It is disappointing that they didn't achieve good "a posteriori" skill, but this is hard to do, and I appreciated their discussion of the issues. I recommended the paper be accepted, but I would do have some comments:"*

We thank the referee for these kind comments. We share his disappointment about the a posteriori instability, and we hope our discussion about it helps the field to eventually overcome it.

*"1. The introduction has a comprehensive literature review of ML for turbulence closures in the CFD/engineering community, but could also mention similar work in the oceanography/atmospheric science on e.g. geostrophic turbulence (Bolton and Zanna, 2019), boundary layer parameterization (McGibbon and Bretherton, 2019), and "fullphysics" parameterization (Brenowitz and Bretherton, 2019; Rasp et. al. 2018; Yuval and O'Gorman 2020). These papers all formulate similar coarse-graining problems, and I think readers would enjoy seeing the connections."*

We agree our introduction could benefit from some additional references, as the referee suggests, to similar work previously done in other geoscientific disciplines, such that the present interesting cross-disciplinary connections are better highlighted. We will therefore include the suggested references.

*Edit 14-4-2021: We included the suggested references at lines 47-49.*

*"2. The question of "a posteriori" and "a priori" errors is a key unsolved problem in the literature. Overall this study diligently accounted for the spatial discretization of the LES, but did not handle the time discretization. For example, would decreasing the time step or using a different time stepper stabilize the "a posteriori" simulations, or is the ML problem they have posed intrinsically unstable?"*

This is a good point raised by the referee. In general though, errors in Navier-Stokes solvers are dominated by errors in the spatial discretizations, due to the strong time constraints imposed by the advection and diffusion. Furthermore,

our time discretization scheme has an inherent damping making the energy decay over time (van Heerwaarden et al., 2017). Hence, increasing errors over time are highly likely from other sources than the time integration itself.

In the description of our unique filtering procedure, we will additionally mention this point.

*Edit 14-4-2021: We included this point at lines 153-155.*

**3.2 Specific comments**

*"L55: "a posteriori tests*

*This jargon has diverged somewhat in the turbulence and weather/climate modeling communities. Climate modelers frequently use the terms "online" and "offline" to denote a posteriori and a priori validations, respectively. It would be good to point this out for the geoscience reader."*

Agree. We will point out the alternative offline/online denotation in the introduction, abstract, and section headers. We will stick to a priori/a posteriori as our main denotation though, as that is the most common one in SGS modelling.

*Edit 14-4-2021: We now highlight the alternative offline/online denotation in the section headers and at lines 53-54.*

*"L95: "Friction Reynold's number"*

*I assume this refers to the Reynolds number arising from molecular viscosity, but it would be clearer if it were defined in terms of the geometry, viscocity etc in Eq 1. Also, it should be pointed out that atmospheric flows have orders of magnitude higher Reynold's numbers than Re=590."*

We agree the current formulation is somewhat confusing for the broader geoscientific community. We will add a note explaining that the friction Reynolds number $Re_\tau$ is a standard measure in studies of wall-bounded turbulence defined based on the friction velocity.
Although $Re_\tau$ is significantly lower than the conventional Reynolds number, the $Re_\tau$ of the selected test case is indeed still lower than for realistic atmospheric flows. We will therefore mention more explicitly in the text, the lower friction Reynolds number compared to reality.

*Edit 14-4-2021: We now mention this at lines 187-188.*

*"L185 (Fig 2). Please define "friction velocity"."*

We will add a brief explanation about the meaning of the "friction velocity" when mentioning the "friction Reynolds number" for the first time (see previous response).

*Edit 14-4-2021: We added an explanation at lines 184-188.*

*"L325. How important was this preferential sampling in terms of quantitative performance?"*

The preferential sampling was mainly important to improve the predictions close to the walls, compared to the predictions in the middle of the channel. In other words, the preferential sampling did not quantitatively improve the prediction accuracy averaged over the whole flow, but rather minimized the difference in prediction performance with channel height. This is desirable, as subgrid models usually matter most close to the walls, where the physics are very different than in the middle of the channel. We will further clarify these points in the corresponding sections.

*Edit 14-4-2021: We clarified the explanations at lines 264-268 and 393-395.*

*"Figure 8. The Smagorinsky scheme does dramatically worse. I am concerned this is a straw-man comparison. Is Smagorinsky a state-of-the-art baseline for this problem?"*

Yes and no. We used the Smagorinsky model because it is one of the most commonly used SGS models in actual large-eddy simulations, despite that it is well-known to perform poorly in a priori tests. We do note that there are other more advanced traditional SGS models that usually perform (somewhat) better in a priori tests (e.g. scale-dependent dynamical SGS models or scale-similarity SGS models).

However, what is in our case crucial, is that we included numerical errors in the definition of $\tau_{ij}$. The Smagorinsky SGS model, together with the other traditional SGS models, do not compensate for these errors. This is an important reason why also other, more advanced SGS models cannot be expected to perform well in the a priori tests presented in this manuscript.

We will revise the a priori test section to emphasize further these points, and highlight that the ANN primarily has potential because of its ability to compensate for numerical errors that other traditional SGS models (like Smagorinsky) do not compensate for.

*Edit 14-4-2021: We emphasize this point now in particular at lines 418-422 and 433-436.*

*"L415. These issues were likely exacerbated by the growing need for the ANN to extrapolate beyond its training state once the simulation started deviating from the physical solution.*

*Many ML-assisted weather models show identical problems. These spiraling errors are can be revealed by normal mode analysis of the underlying linearized ML-fluid system (see Brenowitz, et. al. 2020). It would be fascinating to see this technique applied to NNs here in some future paper."*

We thank the referee for this interesting suggestion. We will keep it in mind for any possible future work on the observed a posteriori instability.

**3.3 Technical corrections**

*"L83: "instantaneuous" is misspelled"*

Indeed. We will correct it.

*Edit 14-4-2021: We now corrected the spelling error at line 90.*

*"Figure 3. This graphic is busy, but I understood what they meant from the text."*

We understand that this figure, in its current form, looks busy and contains a lot of information. Given though that it was clear in combination with the text, we do opt to retain the figure in its current form, as the 3d-visualization does nicely allows us to indicate the calculated surface integrals. We will instead rephrase the caption of the figure, such that the figure hopefully becomes more clear independent from the main text.

*Edit 14-4-2021: In shortening the manuscript, we eventually removed the figure corresponding to this comment.*

*"Figure 9. This graphic has small text and shows some over-plotting in Fig 9a. I suggest further reducing the transparency of the markers or using a hexbin plot"*

Agree. We will make the labels larger, and we will change it to a hexbin plot.

*Edit 14-4-2021: We show the corresponding revised plot in Fig. 8 and 9.*

*"Line 340. This section header needs a preposition e.g. "Comparison \*to\* Smagorinsky""*

Agree. We will change the header to "Comparison to Smagorinsky".

*Edit 14-4-2021: In the restructured, revised manuscript we eventually renamed this section to 'Single horizontal resolution Smagorinsky performance'.*

**References**

Beck, A., Flad, D., and Munz, C.: Deep neural networks for data-driven LES closure models, Journal of Computational Physics, 398, 108 910, https://doi.org/10.1016/j.jcp.2019.108910, 2019.

Brenowitz, N. D. and Bretherton, C. S.: Spatially extended tests of a neural network parametrization trained by coarse-graining, Journal of Advances in Modeling Earth Systems, 11, 2728–2744, https://doi.org/10.1029/2019MS001711, 2019.

Guan, Y., Chattopadhyay, A., Subel, A., and Hassanzadeh, P.: Stable a posteriori LES of 2D turbulence using convolutional neural networks: Backscattering analysis and generalization to higher Re via transfer learning, arXiv preprint arXiv:2102.11400, https://arxiv.org/pdf/2102.11400.pdf, 2021.

Maulik, R., San, O., Rasheed, A., and Vedula, P.: Subgrid modelling for two-dimensional turbulence using neural networks, Journal of Fluid Mechanics, 858, 122–144, https://doi.org/10.1017/jfm.2018.770, 2019.

Park, J. and Choi, H.: Toward neural-network-based large eddy simulation: application to turbulent channel flow, Journal of Fluid Mechanics, 914, https://doi.org/10.1017/jfm.2020.931, 2021.

Rasp, S.: Coupled online learning as a way to tackle instabilities and biases in neural network parameterizations: general algorithms and Lorenz 96 case study (v1.0), Geoscientific Model Development, 13, 2185–2185, https://doi.org/10.5194/gmd-13-2185-2020, 2020.

van Heerwaarden, C. C., van Stratum, B. J. H., Heus, T., Gibbs, J. A., Fedorovich, E., and Mellado, J. P.: MicroHH 1.0: a computational fluid dynamics code for direct numerical simulation and large-eddy simulation of atmospheric boundary layer flows, Geoscientific Model Development, 10, 3145–3165, https://doi.org/10.5194/gmd-10-3145-2017, 2017.

Wang, Z., Luo, K., Li, D., Tan, J., and Fan, J.: Investigations of data-driven closure for subgrid-scale stress in large-eddy simulation, Physics of Fluids, 30, 125 101, https://doi.org/10.1063/1.5054835, 2018.

Xie, C., Wang, J., Li, K., and Ma, C.: Artificial neural network approach to large-eddy simulation of compressible isotropic turbulence, Physical Review E, 99, 053 113, https://doi.org/10.1103/PhysRevE.99.053113, 2019.

Yang, X., Zafar, S., Wang, J., and Xiao, H.: Predictive large-eddy-simulation wall modeling via physics-informed neural networks, Physical Review Fluids, 4, 034 602, https://doi.org/10.1103/PhysRevFluids.4.034602, 2019.

Zhou, Z., He, G., Wang, S., and Jin, G.: Subgrid-scale model for large-eddy simulation of isotropic turbulent flows using an artificial neural network, Computers & Fluids, 195, 104 319, https://doi.org/10.1016/j.compfluid.2019.104319, 2019.